



# Unraveling a black box: An open-source methodology for the field calibration of small air quality sensors

Seán Schmitz[1], Sherry Towers[1], Guillermo Villena[2], Alexandre Caseiro[1], Robert Wegener[3], Dieter Klemp[3], Ines Langer[4], Fred Meier[5], and Erika von Schneidemesser[1]

[1]Institute for Advanced Sustainability Studies e. V. (IASS), Berliner Strasse 130, 14467 Potsdam, Germany
[2]Bergische Universität Wuppertal, Physikalische und Theoretische Chemie/FK4, Gaussstrasse 20, 42119 Wuppertal, Germany
[3]Forschungszentrum Jülich GmbH, Institute of Energy and Climate Research, IEK8: Troposphere, 52425 Jülich, Germany, Germany
[4]Freie Universität Berlin, Institut für Meteorologie, Carl-Heinrich-Becker Weg 6-10, 12165 Berlin, Germany
[5]Chair of Climatology, Institute of Ecology, Technische Universität Berlin, Rothenburgstraße 12, D-12165 Berlin, Germany

*Correspondence to*: Seán Schmitz (sean.schmitz@iass-potsdam.de)

**Abstract.** The last two decades have seen substantial technological advances in the development of low-cost air pollution instruments using small sensors. While their use continues to spread across the field of atmospheric chemistry, the air quality monitoring community, as well as for commercial and private use, challenges remain in ensuring data quality and comparability of calibration methods. This study introduces a seven-step methodology for the field calibration of low-cost sensors using reference instrumentation with user-friendly guidelines, open access code, and a discussion of common barriers to such an approach. The methodology has been developed and is applicable for gas-phase pollutants, such as for the measurement of nitrogen dioxide ($NO_2$) or ozone ($O_3$). A full example of the application of this methodology to a case study in an urban environment using both Multiple Linear Regression (MLR) and the Random Forest (RF) machine-learning technique is presented with relevant R code provided, including error estimation. In this case, we have applied it to the calibration of metal oxide gas-phase sensors (MOS). Results reiterate previous findings that MLR and RF are similarly accurate, though with differing limitations. The methodology presented here goes a step further than most studies by including explicit, transparent steps for addressing model selection, validation, and tuning, as well as addressing the common issues of autocorrelation and multicollinearity. We also highlight the need for standardized reporting of methods for data cleaning and flagging, model selection and tuning, and model metrics. In the absence of a standardized methodology for the calibration of low-cost sensors, we suggest a number of best practices for future studies using low-cost sensors to ensure greater comparability of research.



## 1. Introduction

Air pollution remains a leading cause of premature death globally (Landrigan et al., 2018). The recent trend in air pollution
research of using low-cost sensors (LCSs) to measure common gas-phase and particulate air pollutants (e.g. CO, NOx, O3,
PM) is an attempt to close gaps in our understanding of air pollution and make its measurement cheaper, widespread, and
more accessible (Kumar et al., 2015; Lewis et al., 2016; Lewis et al., 2018). The development of these new technologies
represents a paradigm shift that has opened up air pollution monitoring to a much wider audience (Morawska et al., 2018;
Snyder et al., 2013). In recent years, LCSs have been used to develop or supplement existing air pollution monitoring
networks to provide higher spatial resolution (e.g. CitiSense, U.S. EPA Village Green), as well as in a citizen science
contexts to report on and share information about air quality (e.g. AirVisual, Purple Air) (Morawska et al., 2018; Muller et
al., 2015). Projects like these are a promising step towards empowering citizens with greater knowledge of their local air
quality.

However, as there are myriad commercially available LCSs that use a variety of sensors and have substantial differences in
quality, standardizing their application remains challenging and urgent (Karagulian et al., 2019). In measuring gas-phase
pollutants, for example, metal oxide sensors (MOS) and electrochemical sensors (EC) are often used which have different
limits of detection and cross-sensitivities that need to be taken into account (Lewis et al., 2016; Lewis et al., 2018; Rai et al.,
2017). Under ambient conditions, the performance of these two sensor types varies substantially, with some studies reporting
moderate to good agreement with concentrations measured by reference instrumentation, whereas others find very poor
agreement (Lewis et al., 2018). A further challenge is that many LCSs are in the form of small sensor systems[1] sold as ready-
to-use products to customers, most often using a "black box" proprietary calibration algorithm for producing concentrations
which, along with raw data, is not publicly available (Karagulian et al., 2019). Furthermore, a wide range of calibration
techniques have been applied to LCSs in field studies, but lack uniformity in metrics used, experimental setup, reference
equipment, and environmental conditions, making it difficult to draw conclusions about their overall performance
(Karagulian et al., 2019; Rai et al., 2017).

In general, pairwise reference calibration has been done on an individual sensor system basis as well as a sensor system
cluster basis, also known as "sensor fusion" (Barcelo-Ordinas et al., 2019). The former tends to be more accurate but
becomes logistically and computationally intensive for large numbers of LCSs and is more sensitive to sensor decay and
medium-scale drift. The latter has been shown to be effective at calibrating groups of sensors when using the median sensor
signal of a co-located cluster of sensors to develop a single calibration model applicable to all sensors (Smith et al., 2017;
Smith et al., 2019). Using a cluster-based approach has been shown to produce calibration factors that may be more robust

---

[1] In this case "sensor" and "LCS" refer to the sensor components which react chemically with various air pollutants, whereas
"sensor system" refers to the complete device, including sensors, housing unit, data storage, etc.



over longer time frames, but have higher margins of error for individual sensors. Both methods have their advantages and
disadvantages that must be balanced based on the desired application for the sensor systems. Further methods for calibration
beyond pairwise reference calibration include node-to-node calibration (Kizel et al., 2018) or proxy calibration (Miskell et
al., 2018).

Previous research has used linear regression, multiple linear regression (MLR), and machine-learning techniques such as
random forest (RF), artificial neural networks (ANN), and support vector regression (SVR) to calibrate LCSs with reference
instrumentation for gas-phase pollutants. Here too, there is a lack of standardization, as MLR, RF, ANN, and SVR have all
been found to be the most accurate method across various studies (Bigi et al., 2018; Cordero et al., 2018; Hagan et al., 2018;
Karagulian et al., 2019; Lewis et al., 2016; Malings et al., 2019; Smith et al., 2019; Zimmerman et al., 2018). Only linear
regression has been consistently identified as an unsuitable model, largely because it fails to take into account cross-
sensitivities and environmental influences on sensor functioning and because sensors responses are often non-linear. For this
same reason, nonparametric methods such as the aforementioned machine-learning techniques tend to be more accurate, as
they are better at modelling non-linear sensor responses while being able to better take into account interferences in sensor
functioning (Barcelo-Ordinas et al., 2019; Karagulian et al., 2019). However, it must be said that any of these statistical
methods can be applied as long as they properly account for autocorrelation, multicollinearity, and non-linearity in the data
with relevant transformations.

There are several key issues with previous work on calibrating LCSs that must be acknowledged. First, the metrics used to
report model suitability vary substantially. Karagulian et al. (2019) found in their comprehensive review of the LCS
literature that only the coefficient of determination ($R^2$) was applicable for cross-comparison of all studies. While this metric
can be useful in measuring the agreement between LCS data and reference measurements, it does not give a sense of the
model error. Future studies should, at a minimum, report $R^2$, root mean square error (RMSE), and mean average error
(MAE), when discussing calibration performance (Barcelo-Ordinas et al., 2019; Karagulian et al., 2019). Second, while there
are a host of studies that calibrate LCSs with MLR or machine-learning techniques, the associated model selection,
validation, and tuning methods are rarely reported. The latter of these is especially important for machine-learning (ML)
techniques with many tuning parameters, where the problem of over-fitting is more common. Some studies do report steps
for model validation (Hagan et al., 2018; Spinelle et al., 2015; Zimmerman et al., 2018) or model tuning (Bigi et al., 2018;
Spinelle et al., 2015), but they do not go into depth as to how these were determined or optimized. Especially with "black
box" techniques such as ANN, SVM, or RF, reporting steps taken to validate the model and optimize parameters is crucial to
ensuring consistency among studies. Last, the issues of multicollinearity and autocorrelation, which are common among LCS
time series data and of substantial importance when using MLR, are rarely addressed. If at all mentioned, they are referred to
as being better handled by non-linear ML techniques such as SVM or RF (Bigi et al., 2018) or as potentially obscuring the





statistical significance of models (Masiol et al., 2018). This study seeks to take a step forward in ensuring these issues are addressed in future LCS calibration studies.

In the absence of a standardized calibration methodology, the ever-growing body of LCS literature will continue to be largely incomparable, with research running in parallel using varied methods. Though several comprehensive reviews of LCSs have been completed which establish helpful guidelines for their use (Lewis et al., 2018; Williams et al., 2014), best practices for calibration with reference instruments that should be undertaken in any field deployment were not specifically reported. More recently, Barcelo-Ordinas et al. (2019) published an extensive study on the calibration of LCSs, including

some general calibration guidelines. While these are a helpful guide for calibration methodologies, they lack important details on the post-processing of data during the model-building process. This study seeks to expand upon this work and specifically address the standardization of individual pairwise calibration of LCSs with reference instrumentation by presenting user-friendly guidelines, open access code, and a discussion of common barriers to field calibration. With the publication of this step-by-step methodology for the statistical calibration of LCSs, we hope to establish a framework from

which calibration methods can be better compared.

## 2. Methods

The following section outlines a methodology for the deployment and field calibration of LCSs for the measurement of gas-phase pollutants. While it was applied here to metal oxide LCS, the methodology is also equally as applicable to

electrochemical LCS. First, some key considerations for the experimental deployment of small sensor systems will be discussed. Second, a 7-step statistical calibration methodology for the post-processing of data will be described. Last, an example of the use of this methodology, both for deployment and calibration, using data collected during a measurement campaign in 2017 and 2018, is provided (Section 3).

For this methodology, it is important to first establish under which circumstances the following steps would apply. This is a reference-based pairwise method for the individual calibration of small sensors and therefore the user will need to have access to reference instrumentation with which the small sensor systems can be co-located, whether their own or in collaboration with e.g. a city monitoring network. This makes the methodology inapplicable for individual users in a citizen science context that may not have access to reference instrumentation. These reference instruments should adhere to

standardized guidelines on accuracy (i.e. EU Air Quality Directive (2008/50/EC), U.S. National Ambient Air Quality Standards (NAAQS)). A co-location in this sense refers to the installation of the small sensor systems in the close vicinity (ca. 1-3 meters) of the reference instruments, so that they receive the same parcels of air. This paper focuses on the usage of field (i.e. in-situ) co-locations in calibrating small sensor systems. If access to reference data or the raw small sensor data is not possible, then this methodology cannot be applied.




### 2.1 Key considerations for the experimental deployment of small sensor systems

When calibrating small sensor systems, the experimental deployment and co-location of devices is a key step with several important considerations that must be accounted for. First, the co-location with reference instrumentation should ideally occur at the same test site where the small sensor systems are to be deployed. If unfeasible for logistical reasons, an analogue

site should be selected. Criteria for analogue selection entail similar characteristics as those for test site selection. The analogue site should: 1) have similar sources and ranges of concentrations of air pollutants as the test site; 2) experience comparable meteorological conditions and similar circulation dynamics; and 3) be physically located in the same region as the test site. While it is unlikely that there will be a perfect analogue site, any field-calibration should take these criteria into consideration in order to enhance validity of experimental results.


Second, the frequency and timing of co-locations should reflect site-specific variations in meteorological conditions. Generally, these should be done often enough so that co-location data cover similar ranges of meteorological conditions and concentrations of air pollutants as the experimental data, but not so often that there is a concomitant loss of experimental data. A rule-of-thumb for long-term experiments (>6 months) in temperate seasonal environments is a 2 week co-location

every 2-3 months. For short- to medium-term experiments, a 2 week co-location before and after and perhaps one in-between, depending on changes in meteorological conditions, should suffice. Regular co-location allows for the establishment of datasets that cover not only changes in meteorology, but also sensor functioning and health. If these considerations are taken into account during the experimental deployment, the likelihood that these datasets will be of good quality for use in statistical calibration will be higher.


### 2.2 The 7-step statistical calibration method

Raw data from small sensor systems, if treated and transformed properly, can provide informative air pollutant concentrations. This treatment must, however, be rigorous if the resultant concentrations are to be used in further analysis. What follows in this section is a general description of a seven step methodology for the post-processing and calibration of

LCS data. Multiple Linear Regression (MLR) and Random Forest (RF) were selected as calibration methods to be used in this methodology, although it can be generally applied to other regression or machine learning methods. Information on the functions and packages from the open-source R statistical software program (R Core Team, 2019) used in this methodology is provided for each step. This information and the code can be found in the open source repository Zenodo (see below for DOI).




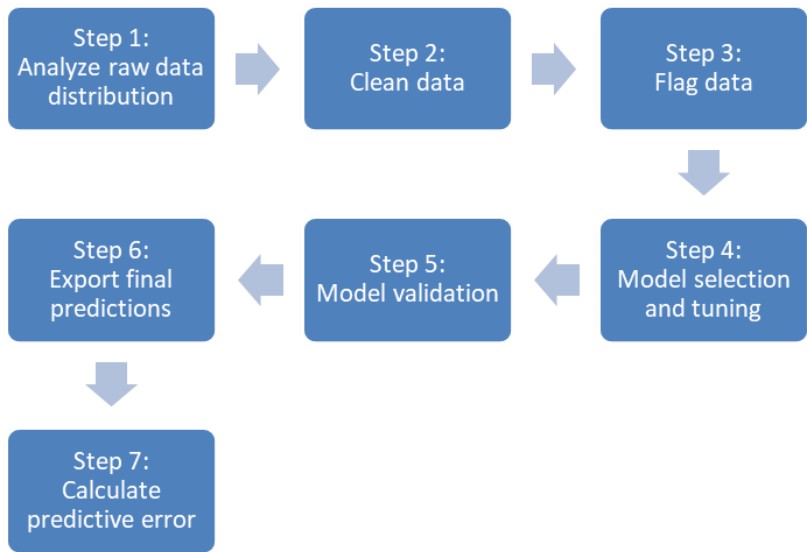

**Figure 1. Schematic representation of the seven-step calibration method for processing small sensor system data.**

### 2.2.1 Step 1: Analyze and understand raw data distribution

The first step is to gain a general understanding of the data. Specifically, establishing an overview of data distributions and potential data quality issues (data gaps, presence of outliers, changes in baselines, etc.) is helpful for identifying problems and solutions during calibration. It should also be checked that all associated metadata are available for all datasets.

In this study, all variables that were to be used in model selection were assessed in this step. For example, the distributions of
the reference concentrations, small sensor system raw data, and meteorological variables from the co-location and experimental datasets were analyzed. Meteorological variables including temperature, relative humidity, and wind speed and direction across the co-location and experimental datasets were compared. Additional variables that could be considered but were not analyzed here include precipitation, boundary layer height, and insolation, among others. A visual assessment of these data using histograms, violin plots, and time series plots was conducted. This step provided information about the
structure of each available co-location dataset and the experimental dataset crucial to decision-making in later steps.

### 2.2.2 Step 2: Data cleaning

Next, the datasets should be cleaned of erroneous outliers and unreliable data. This step is crucial, as outliers can have a particularly strong effect on calibration models and especially so on linear regression models.


To accomplish this, the time series plots generated in step 1 were first used to visually evaluate the data. Sequence outliers resultant from sensor warm-up time or sensor malfunctioning were identified and removed using an automated function. Next, an algorithm was tested, trained, and implemented that uses a simple z-test with a running mean and standard deviation



to detect point outliers resultant from instrument measurement error. Tests of normality with datasets greater than 50 points are irrelevant in determining whether parametric tests can be used or not (Ghasemi and Zahediasl, 2012). Analysis of the data in this study revealed the same, as data segments of less than 30 points consistently passed the Shapiro-Wilk test, but with progressively larger data segments, more and more of the data failed the test. Therefore, it was assumed that the data aligned enough with the normal distribution for this test to apply. The size of the moving time frame from which the running mean and standard deviation were calculated and the z-score threshold used to designate 'outlierness' were tested and optimized. Durations of 1, 2, 5, 10, 30, 60, 120, and 300 minutes were considered for the moving window and thresholds of 3, 4, 5, and 6 were tested. This was done for each variable individually. Other outlier detection functions using AutoRegressive Integrated Moving Average (ARIMA) and Median Absolute Deviation (MAD) were tested and were found to be inappropriate for this data.

### 2.2.3 Step 3: Flag data for further scrutiny

Experimental data outside the range of the co-location data should be flagged as they may be less reliably predicted than those which are in-range and should be given a higher level of uncertainty (Smith et al., 2019). Flagging such data points strikes a balance between removing them from the analysis and highlighting their associated uncertainty.

Once flagged, these data points were treated differently in later analysis (Section 3.4). Similarly, co-location data outside the range of the experimental data received a flag. During the model selection process, these flags were used to remove data that may serve to bias the model. While this may seem unnecessary, if the experimental range of environmental conditions is much smaller than those of the co-locations, it could be that using a smaller, more comparable range of co-location data is more suitable for model selection. This is data and model dependent, however, and was therefore tested in Step 6.

### 2.2.4 Step 4: Model selection and tuning

Model selection and tuning is a seldom-reported step that is vital in ensuring the calibration model is suitable for use. Rigorously scrutinizing a variety of potential models and optimizing their parameters provides reproducible justification for the final model selected. This is particularly important for machine-learning techniques which can have a wide array of parameters for tuning model performance. Furthermore, appropriate methods used in model selection ensure that problems of multicollinearity and autocorrelation can be corrected for, as superfluous predictors suffering from these issues will be identified and removed. Before building and selecting potential models, the relationships between predictors and response variable, including potential transformations, must be determined. This is important for linear regression models, but is not relevant for ML techniques which do not take these transformations into account. Often the sensor specifications indicate what type of transformation (exponential, log-linear, etc.) may be necessary.



In this case, log transformations were recommended for the MOS sensors used, but were cross-checked with other common transformations including: log-log, square-root, and inverse. Model selection proceeded through backwards selection using the coefficient of determination (R2), root mean squared error (RMSE), and the Akaike Information Criterion (AIC) for
MLR or Variable Importance (VI) for RF as criteria. To determine the best models, the training data set was broken up into smaller sets by using a moving window of four days to train the models and the fifth day to test. The models with the best average RMSE over the various fifth day predictions were selected.

For RF the model parameters of mtry (the number of randomly selected variables at each node), min.node.size (the minimum
number of data points in the final node), and splitrule (the method by which data are split at each node) were optimized by testing various combinations and selecting the most accurate in terms of RMSE, with data split in the same manner as for MLR. Subsequently, measures of AIC for the regression model and VI for the random forest model were assessed to determine which predictors should remain in the model. For MLR, this involved the repeated bootstrapping of the training set combined with stepwise selection, using the AIC to robustly determine predictor inclusion. The models were then finally
tested on the test subset and assessed using RMSE and R2. The most accurate MLR and RF models were then sent to the next step for validation.

### 2.2.5 Step 5: Model validation

Model validation is often overlooked but is necessary to ensure that the most accurate model selected is reliable (i.e. has
good predictive power for independent data). While a singular instance of splitting the dataset during the model selection process into training and testing subsets is one method of validating the model, an additional step ensures more rigorous validation.

In this case, to validate the MLR and RF models selected in Step 4, the co-location data was repeatedly split into training and
testing subsets at a ratio of 75/25. This was done by splitting the co-location training set into continuous blocks representing 25% of the training data (in this case 6 days) as test subsets and using the rest of the co-location data to train the model. Using continuous blocks instead of random sampling is necessary to account for the autocorrelation in the data (Carslaw and Taylor, 2009). The accuracy of the final models was then assessed on the continuous blocks using R2, RMSE, and Variable Importance. These metrics were then graphed across all continuous blocks to assess model stability. If the graphs showed
instability across the various folds, Step 4 was repeated and a new model was selected for validation.

### 2.2.6 Step 6: Export final predictions

Once the selected model has been validated, the next step in the process is to export predictions of the experimental data as concentrations. Only co-location data deemed relevant from the Steps 1-3 should be used to train the model, which is then
used to predict experimental concentrations.



In this case, the co-location data was used to train the best MLR and RF models identified in Steps 4 and 5. These models were then applied to the raw experimental data in order to predict final concentrations. The final predictions were then graphed and compared using time plots and histograms.


### 2.2.7 Step 7: Calculate total predictive error

Last, it is vital that overall error and confidence intervals for the predictions are reported in this step. Most models have associated methods for reporting metrics such as standard error which can be used to establish confidence intervals around the predictions. Compounded to this must be the technical error associated with measurements from the reference

instruments. Thus, the overall error should combine technical and statistical error.

In this study, to test the impact of the precision of the reference measurements on model accuracy, the reference NO2 and O3 data were smeared using a normal distribution with each point as the mean and each instrument's measure of imprecision as the standard deviation. This test therefore determined whether the imprecision given by each instrument's specifications

should be factored in to the overall predictive error. This was done over 50 iterations to see how model accuracy responded to shifts in reference concentrations within the margins of error. Co-location data were split 75/25 into a training set and testing set, respectively. In each iteration, separate MLR and RF models for NO2 and O3 were trained; each was trained once with the reference measurements and once with smeared reference measurements. All models were then tested for predictive accuracy on the testing subset, to compare the impact of smeared versus measured reference data on model

performance.

Last, the overall uncertainty was calculated. For the reference instruments, the technical measurement was taken from their specifications. This was added to the overall statistical error, for which the median MAE across all blocks from the model validation step was used. Both the MLR and RF model calculated a measure of standard error, which was compared with the

combined uncertainty measure. The more appropriate of the two was then added to the final predictions from Step 6.

## 3. Example application of the methodology

### 3.1 Small sensor systems used

The small sensor systems used in this example are EarthSense Systems, Ltd. "Zephyr" prototypes[2], henceforth referred to as

"Zephyrs". This term refers to the whole small sensor system including housing, sensors, GPS, etc. Installed within the Zephyr prototypes were a number of Metal Oxide Sensors (MOS) that measure reducing gases, oxidizing gases (used here for detection of nitrogen dioxide), ozone, and ammonia, as well as a meteorological sensor for temperature and relative

---

[2] The Earthsense Zephyrs have since evolved substantially and, as such, this study does not represent current performance or configuration.



humidity; see Table 1 for more on these sensor specifications. For greater detail on the development, functioning, and operation of the sensors housed within these prototypes see Peterson et al. (2017).


**Table 1. Sensors installed within the EarthSense Zephyr prototypes. Table reproduced from Peterson et al. (2017).**

| Gases Measured | Sensor Model | Method of detection | Gas detected and detection limits |
|---|---|---|---|
| Reducing gases | SGX Sensortech MICS-4514 | Redox reaction | CO: 1-1000 ppm<br>$NH_3$: 1-500 ppm<br>$C_2H_5OH$: 10-500 ppm<br>$H_2$: 1-1000 ppm<br>$CH_4$: >1000 ppm |
| Oxidising gases | SGX Sensortech MICS-4514 | Redox reaction | $NO_2$: 0.05-10 ppm<br>H2: 1-1000 ppm |
| Ozone | SGX Sensortech MICS-2614 | Redox reaction | 10-1000 ppb |
| Ammonia | SGX Sensortech MICS-5914 | Redox reaction | $NH_3$: 1-500 ppm<br>$C_2H_5OH$: 10-500 ppm<br>$H_2$: 1-1000 ppm<br>$C_3H_8$: >1000 ppm<br>$C_2H_8(CH_4)_2$: >1000 ppm |
| Temperature and relative humidity | GE Measurement and Control CC2D25 | Polyamide capacitance | Temp.: -40 – 125 °C<br>RH: 0 – 100% |

## 3.2 Reference instruments

The reference instrumentation included a Teledyne Model T-200 NO/NO2/NOx Analyzer and a 2B Technologies, Inc.
Ozone Monitor. These instruments were intercompared with reference instruments – CAPS (Aerodyne, U.S.A.), CLD 770 AL ppt (ECOPHYSICS, Switzerland) and O242M (Environnement S.A., France) – from the Forschungszentrum Jülich as part of the measurement campaign and showed decent agreement for NO2 and good agreement for O3 (see Figures S1-S3 in the supplemental information). Ambient air temperature and relative humidity (Lambrecht, PT100) data one block away from the experimental site were provided by the Free University Berlin for two measurement campaigns (more information
in section 3.2). Wind speed and direction (Campbell Scientific, IRGASON), were measured 10m above the roof of the main building of the Technical University Berlin (TUB) at Campus Charlottenburg, which is located across the street from the experimental site. This site is part of the Urban Climate Observatory (UCO) Berlin operated by the TUB for long-term observations of atmospheric processes in cities (Scherer et al., 2019a).

**3.3 Experimental deployment**

Measurements were conducted in a street canyon on the Charlottenburg Campus of the TUB, on the façade of the Mathematics Building (52° 30' 49.7" N, 13° 19' 34.5" E) as a part of several measurement campaigns of the joint project 'Three-dimensional observation of atmospheric processes in cities' (3DO) (Scherer et al., 2019a), which was part of the



larger research program Urban Climate Under Change [UC]2 (Scherer et al., 2019b). The area directly around the measurement site consists of university buildings with a wide main thoroughfare (Strasse des 17. Juni) that runs from East to West through Berlin (see Figure S4 in the supplementary information). The experiments took place from July 29th – August 28th and from September 20th – October 12th in 2017, and from January 27th – February 23rd in 2018. These occurred during two measurement campaigns which are henceforth referred to as the Summer Campaign (SC), which includes all 2017 measurements, and the Winter Campaign (WC) which includes the 2018 measurements, respectively (Figure 2).

For the field calibration the Zephyrs were co-located with the aforementioned reference instruments at the deployment site. The reference station for co-location was set up in an office on the 6th floor of the Mathematics building on the south facing façade that provided constant power for reference instrumentation and the Zephyrs, as well as space for air inlet tubing to be passed through the windows to the reference instrumentation (Figure 3). The Zephyrs and the air inlets were attached next to each other on the same railing outside the office. This ensured that all instruments were receiving the same parcels of air throughout the co-location. The reference station measurements were continuous throughout the co-locations and the experiments. Five co-locations were conducted in total across the two campaigns. These took place from July 18th – July 27th, August 29th – September 7th and October 14th – October 27th (all in 2017) during the SC and from January 13th – January 24th and February 23rd – March 8th (both in 2018) during the WC. All dates refer to time frames of the data presented, as the first and last days of deployment or co-location were not used owing to different start and end times of installation, as well as sensor warm up times.

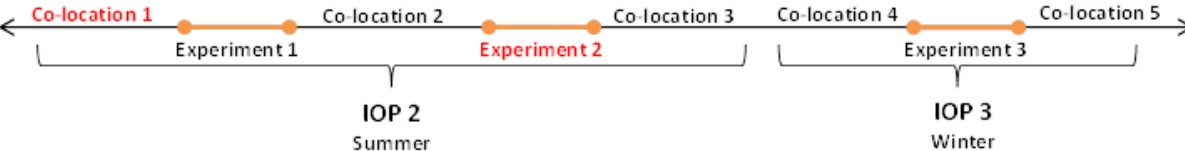

**Figure 2. Timeline of SC and WC depicting the relationship between co-locations and experiments. Due to technical issues of individual instruments, data were unavailable for the segments marked in red.**

This example focuses on Zephyr s71 during the SC and Zephyr s72 during the WC. For the sake of brevity, all graphs and tables included in this section pertain only to the former. Those relevant for the latter can be found in the Supplementary Information. One Zephyr was co-located with reference instrumentation throughout the summer campaign (s71) and winter campaign (s72) measurement campaigns on the 6th floor of the Mathematics building. As such, the statistical models established using the 7-step method could be trained with co-location data and, atypically, assessed for their accuracy using reference concentrations during the entire experimental window. What follows is a thorough description of the application of the seven-step method for calibration.





In order to calibrate the Zephyrs, reference data, meteorological data, and raw data from the Zephyr sensors were used.
Concentrations of NO2 from the Teledyne T200 NOx Analyzer and O3 from the O3-2B Technologies instruments were used as response variables in the models. Ambient temperature (Tamb) and relative humidity (RHamb) data as well as wind speed (ws) and direction (wd) data were tested as predictors in the statistical models. Four variables from the Zephyrs themselves were also tested in the statistical models as predictors: 1) Oxa, a measure of voltage from one MOS sensor used to detect oxidizing substances (in this case NO2); 2) O3a, another measure of voltage from a MOS sensor that detects ozone; 3) a
measure of temperature collected by the Zephyr (Tint); and 4) a measure of relative humidity collected by the Zephyr (RHint). Finally, the binary time-of-day (ToD) variable was created to distinguish between night and day, as the chemistry of the analyzed species changes significantly.

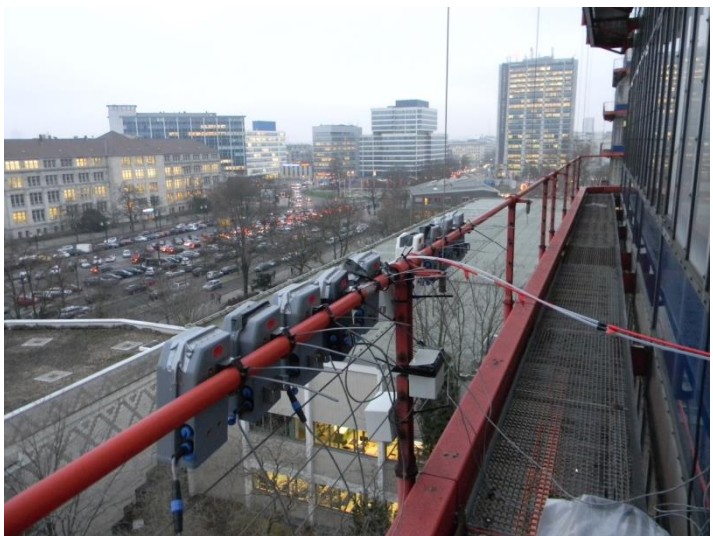

**Figure 3. Set-up of the co-location of the prototype Zephyrs with reference instruments on the 6th floor of the Maths building. The**
**grey units are the Zephyrs and the two inlet tubes connect to the reference devices located inside the office.**

### 3.4 Seven-step calibration of the Zephyrs

The temperature and relative humidity from the Zephyrs (Tint and RHint) reflect the conditions within the sensor system and typically parallel ambient data, however, with an offset. These data are henceforth referred to as "internal" temperature and
relative humidity. Throughout the example, both internal and ambient T and RH are used to assess their comparative influence on model accuracy. This was tested as ambient T and RH from reference instruments are not always available at experimental sites, whereas the internal T and RH of the Zephyrs are always available. The reference and meteorological data had an original time resolution of 1 minute whereas the Zephyr data was collected at a time resolution of 10 seconds. Analysis during the seven-step process was conducted using 5 minute averages except for outlier detection, which was done
at original time resolution.





### 3.4.1 Step 1: Analyze raw data distribution

The distributions of the reference, meteorological, and Zephyr data were first compared between each co-location, the co-locations combined, and the experimental deployment data of Experiment 1. The violin plots of RHamb, Tamb, NO2, and

O3 for co-location 2 (Figure 4) show that the meteorological conditions and pollutant concentrations experienced were quite similar to those of the experiment. The ranges, median values, and the interquartile ranges are quite similar. This is further reflected by the similarity in distributions of both the Zephyr MOS sensor data (Oxa and O3a) and the reference instrument data between the 2nd co-location and the experiment.

By contrast, the distributions of the same variables for the 3rd co-location (Figure 4) are demonstrably different from the other co-location and the experimental data. The ambient temperature and relative humidity conditions were significantly cooler and wetter in the 3rd co-location than during the experiment and the NO2 and O3 concentrations were much higher and lower, respectively. Furthermore, the MOS sensor data in this co-location have a much different median and IQR than the experiment although the overall range is similar.


With both locations combined (Figure 4), the distributions of all variables are representative of the experimental data, but with worse agreement than with co-location 2 alone. These results suggested that the 2nd co-location alone could be the best training set for the model building process. In order to further assess this hypothesis, co-location 2, co-location 3, and a combination of both were used in exporting final model predictions and evaluated using the atypical co-located experimental

data as a "comparison" dataset.





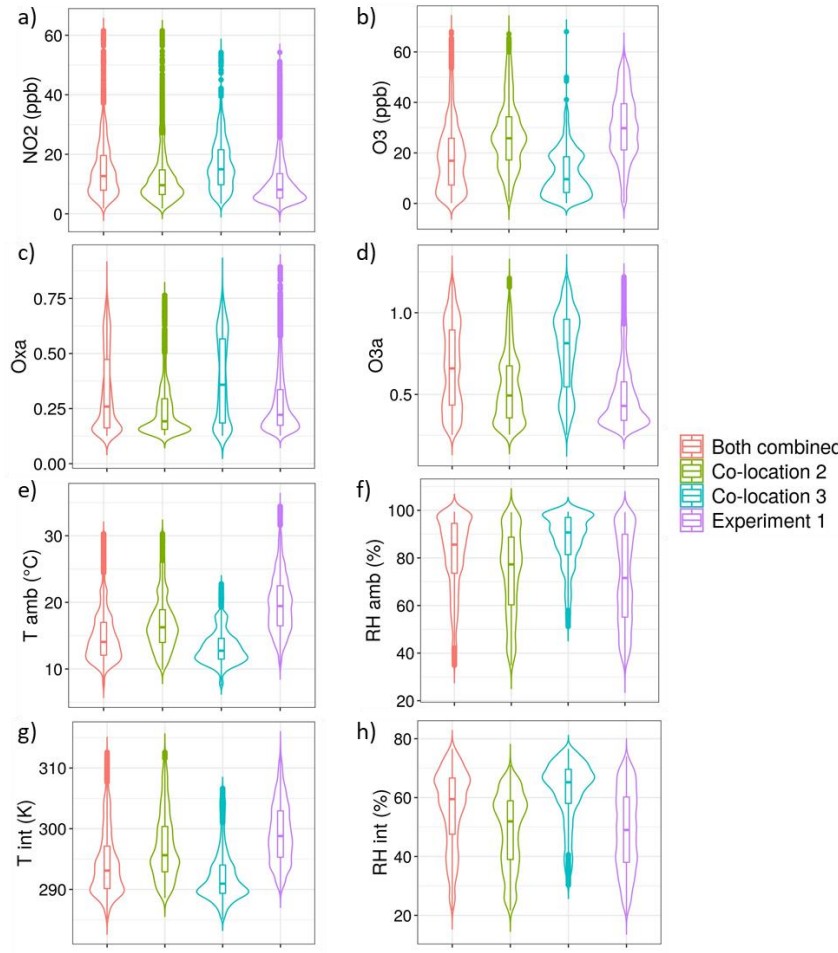

**Figure 4. Violin plots of a) reference NO2, b) reference O3, c) Oxa, d) O3a, e) Tamb, f) RHamb, g) Tint, and h) RHint for co-location 2, co-location 3, both co-locations combined, and the experimental data.**

### 3.4.2 Step 2: Data cleaning

Point outliers were determined using the developed outlier detection function. The threshold and running window parameters were optimized individually for each variable. This was done through visual assessment of points identified as outliers under various parameters, in order to determine if the designation was appropriate. For the reference NO2 and O3 data, using a z-score threshold of five and a running mean calculated with 120 data points (equivalent to two hours of data) was optimal for

identifying true outliers. Using a lower threshold often falsely identified the extremes of normal data spikes as outliers. Figure 5 shows example outliers that were identified using the function described above for the reference data. The reference relative humidity and temperature data provided by the Free University had been pre-processed and as such no outliers were identified in those data.



As part of normal operation, the Zephyrs send logged data via GSM connection every 15 minutes to a database maintained by EarthSense. When this occurs, all metal oxide sensors in the device turn off. The MOS sensors by design, however, run quite hot and require a constant input of power to maintain their temperature. As can be seen in Figure 6, each time the MOS sensors turn off, they need to warm-up again before stabilizing. The time series plots developed in Step 1 were key to identifying and addressing this issue. By developing a function in R that analyzes the MOS sensor data patterns following

time-gaps due to GSM connection, we developed a rule-of-thumb for identifying and removing these data. Analysis of this issue showed that the sensors required two and a half minutes to warm-up and return to normal functionality.

Once the time-gap anomalies were removed from the Zephyr data, the outlier detection function was applied to the four Zephyr variables in original time resolution. As can be seen in Figure 7, outliers were detected for the four Zephyr variables

with a z-score threshold of five and a running mean of 360 data points (equivalent to one hour of data). It is likely that these anomalous data points all result from brief technical failures within the instrument.

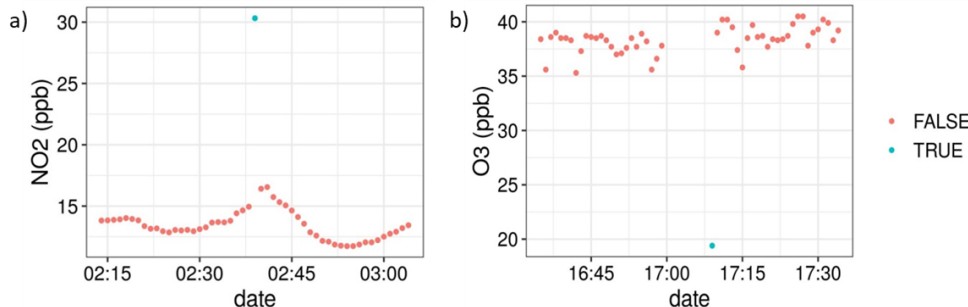

**Figure 5: Examples of outliers detected on reference data using a z-test with running mean for the SC. A value of "TRUE" means the point was deemed an outlier by the z-test.**

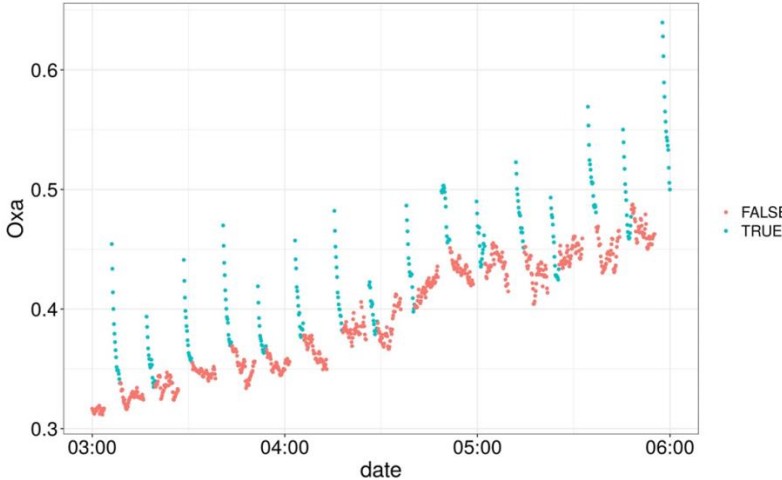


**Figure 6: Example of outliers due to MOS sensor warm-up following a GSM connection of the Zephyrs. A value of "TRUE" indicates the point was included in the 2.5 minute MOS warm-up period.**



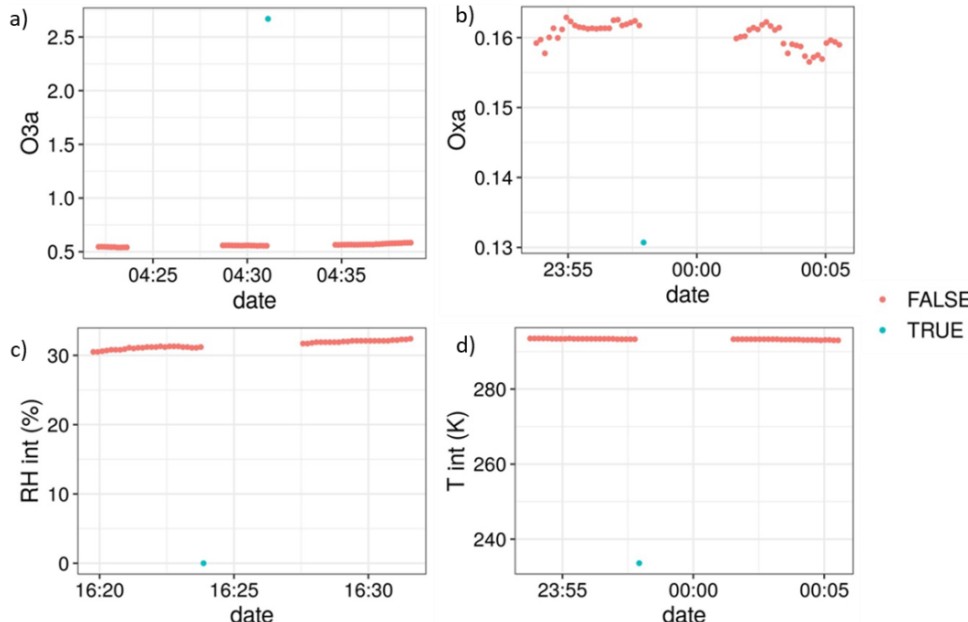

**Figure 7: Examples of outliers detected on Zephyr s71 data using a z-test with running mean for the SC. A value of "TRUE"**
**means the point was deemed an outlier.**

### 3.4.3 Step 3: Flagging the data

Given that the data coverage from the 2nd co-location encompassed most of the experimental data, only a few points during
the experiment were flagged for being out-of-bounds of the 2nd co-location set. As can be seen in Figure 8a, only low NO2
concentrations from the experimental set were flagged. The 3rd co-location experienced a more narrow range of NO2
concentrations, as can be seen in Figure 4 from Step 1. As such, more experimental data points of lower concentrations and
some of high concentrations were flagged for this co-location (Figure 8b).

Similarly, the 2nd co-location dataset received few flags, as most variables had comparable ranges to those of the
experimental dataset. For example, only a few data points in which the internal Zephyr temperature dipped below ~289K
were flagged (Figure 8c). For the 3rd co-location, which was conducted in colder conditions in October, far more data points
were flagged (Figure 8d). This indicated that a larger portion of the 3rd co-location could be unsuitable for use in calibration.
It also proved valuable for later analysis when analyzing the final predicted concentrations of the model in Step 6.


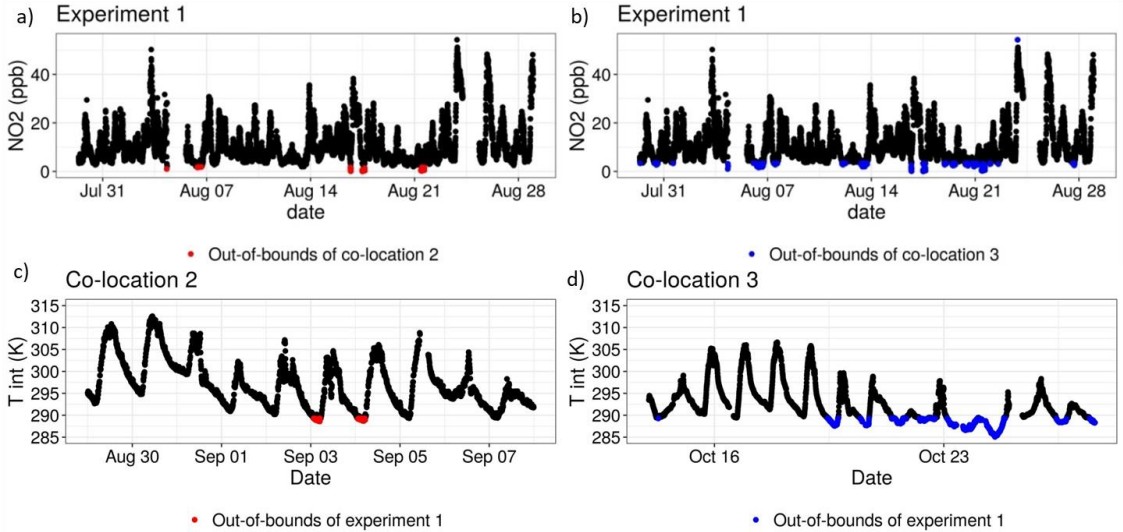

**Figure 8. a) and b) Example time series plots of the experimental data with points out-of-bounds of the 2nd and 3rd co-location flagged, respectively. c) Time series plot of the 2nd co-location with points flagged for being out-of-bounds of the experimental data set. d) Time series plot of the 3rd co-location with points flagged for being out-of-bounds of the experimental data set.**

### 3.4.4 Step 4: Model selection

The results of the model selection process can be seen in Tables 2-5. For readability, these tables reflect a later stage in the process, after which a wide range of other models had already been tested and excluded on the basis of AIC and accuracy metrics. A combination of these metrics was used to designate the "best" models in which RMSE and R2 had a somewhat higher priority than AIC. The most accurate MLR model for predicting NO2 was determined to be one in which Oxa, O3a, RH, and T were included as single terms with interactions between all variables. The relationship between NO2 and Oxa was determined to be logarithmic. For O3 the most accurate MLR model had Oxa, O3a, RH, and T included as single terms with interactions. The relationship between Oxa and O3 was determined to be inverse. However, as the predictive accuracy between this model and one with no transformation of O3 was almost identical, the latter was selected. In such a case, it is best to select the simpler model without transformations. For both NO2 and O3, MLR models using ambient T and RH were consistently more accurate than those using internal T and RH.





**Table 2. Results of the MLR model selection process for NO2. The most accurate model is in bold font. RMSE and MAE are in units of ppb.**

| Formula | $R^2$ | RMSE | MAE | AIC |
|---|---|---|---|---|
| **$NO_2 \sim \log(Oxa) * O3a * RH_{amb} * (1/T_{amb})$** | **0.78** | **4.58** | **3.71** | **26392.56** |
| $NO_2 \sim \log(Oxa) + O3a + RH_{amb} + (1/T_{amb})$ | 0.82 | 4.60 | 3.77 | 26907.25 |
| $NO_2 \sim \log(Oxa) * O3a * RH_{amb} * T_{amb}$ | 0.73 | 5.15 | 4.29 | 25760.14 |
| $NO_2 \sim \log(Oxa) + O3a + RH_{amb} + T_{amb}$ | 0.80 | 4.76 | 3.95 | 26464.38 |
| $NO_2 \sim \log(Oxa) * O3a * RH_{int} * (1/T_{int})$ | 0.70 | 5.60 | 4.54 | 27408.47 |
| $NO_2 \sim \log(Oxa) + O3a + RH_{int} + (1/T_{int})$ | 0.75 | 5.38 | 4.53 | 28115.86 |
| $NO_2 \sim \log(Oxa) * O3a * RH_{int} * T_{int}$ | 0.65 | 6.00 | 4.81 | 26516.68 |
| $NO_2 \sim \log(Oxa) + O3a + RH_{int} + T_{int}$ | 0.74 | 5.25 | 4.42 | 27892.97 |

**Table 3. Results of the MLR model selection process for O3. The most accurate model is in bold font. RMSE and MAE are in units of ppb.**

| Formula | $R^2$ | RMSE | MAE | AIC |
|---|---|---|---|---|
| $O_3 \sim \log(Oxa) * O3a * RH_{amb} * T_{amb}$ | 0.83 | 3.81 | 2.72 | 23930.73 |
| $O_3 \sim \log(Oxa) + O3a + RH_{amb} + T_{amb}$ | 0.87 | 3.26 | 2.58 | 25191.45 |
| **$O_3 \sim \log(Oxa) * O3a * (1/RH_{amb}) * T_{amb}$** | **0.86** | **3.25** | **2.44** | **24214.79** |
| $O_3 \sim \log(Oxa) + O3a + (1/RH_{amb}) + T_{amb}$ | 0.88 | 3.31 | 2.61 | 25255.67 |
| $O_3 \sim \log(Oxa) * O3a * RH_{int} * T_{int}$ | 0.77 | 4.39 | 3.37 | 26151.18 |
| $O_3 \sim \log(Oxa) + O3a + RH_{int} + T_{int}$ | 0.80 | 3.91 | 3.18 | 28106.65 |
| $O_3 \sim \log(Oxa) * O3a * (1/RH_{int}) * T_{int}$ | 0.78 | 4.42 | 3.38 | 26377.66 |
| $O_3 \sim \log(Oxa) + O3a + (1/RH_{int}) + T_{int}$ | 0.79 | 4.17 | 3.43 | 28213.73 |

**Table 4. Results of the RF model selection process for NO2. Min.node.size and split rule were optimized in a previous step not shown here for brevity and are therefore constant. RMSE and MAE are in units of ppb.**

| Formula | mtry | min. node.size | Split rule | $R^2$ | RMSE | MAE |
|---|---|---|---|---|---|---|
| $NO_2 \sim Oxa + O3a + RH_{amb} + T_{amb} + ToD + wd + ws$ | 7 | 5 | extratrees | 0.69 | 5.18 | 4.15 |
| $NO_2 \sim Oxa + O3a + RH_{amb} + T_{amb} + ToD + wd$ | 6 | 5 | extratrees | 0.68 | 5.28 | 4.27 |
| $NO_2 \sim Oxa + O3a + RH_{amb} + T_{amb} + ToD$ | 5 | 5 | extratrees | 0.68 | 5.31 | 4.31 |
| $NO_2 \sim Oxa + O3a + RH_{amb} + T_{amb}$ | 4 | 5 | extratrees | 0.72 | 4.89 | 3.99 |
| **$NO_2 \sim Oxa + O3a + RH_{amb}$** | **2** | **5** | **extratrees** | **0.74** | **4.52** | **3.60** |
| $NO_2 \sim Oxa + O3a + T_{amb}$ | 2 | 5 | extratrees | 0.68 | 5.77 | 4.78 |
| $NO_2 \sim Oxa + O3a$ | 2 | 5 | extratrees | 0.70 | 5.02 | 4.04 |
| $NO_2 \sim Oxa + O3a + RH_{int} + T_{int} + ToD + wd + ws$ | 7 | 5 | extratrees | 0.60 | 5.93 | 4.73 |
| $NO_2 \sim Oxa + O3a + RH_{int} + T_{int} + ToD + wd$ | 6 | 5 | extratrees | 0.61 | 6.03 | 4.81 |
| $NO_2 \sim Oxa + O3a + RH_{int} + T_{int} + ToD$ | 5 | 5 | extratrees | 0.58 | 6.48 | 5.10 |
| $NO_2 \sim Oxa + O3a + RH_{int} + T_{int}$ | 3 | 5 | extratrees | 0.62 | 6.15 | 4.98 |
| $NO_2 \sim Oxa + O3a + RH_{int}$ | 2 | 5 | extratrees | 0.62 | 6.12 | 4.92 |
| $NO_2 \sim Oxa + O3a + T_{int}$ | 2 | 5 | extratrees | 0.68 | 5.77 | 4.78 |






**Table 5. Results of the RF model selection process for O3. Min.node.size and split rule were optimized in a previous step not shown here for brevity and are therefore constant. RMSE and MAE are in units of ppb.**

| Formula | mtry | min. node.size | Split rule | $R^2$ | RMSE | MAE |
|---|---|---|---|---|---|---|
| $O_3$ ~ Oxa + O3a + $RH_{amb}$ + $T_{amb}$ + ToD + wd + ws | 4 | 5 | extratrees | 0.86 | 3.29 | 2.55 |
| $O_3$ ~ Oxa + O3a + $RH_{amb}$ + $T_{amb}$ + ToD + wd | 4 | 5 | extratrees | 0.86 | 3.21 | 2.42 |
| $O_3$ ~ Oxa + O3a + $RH_{amb}$ + $T_{amb}$ + ToD | 2 | 5 | extratrees | 0.86 | 3.20 | 2.41 |
| $O_3$ ~ Oxa + O3a + $RH_{amb}$ + $T_{amb}$ | 2 | 5 | extratrees | 0.86 | 3.20 | 2.50 |
| $O_3$ ~ Oxa + O3a + $RH_{amb}$ | 2 | 5 | extratrees | 0.84 | 3.83 | 3.05 |
| **$O_3$ ~ Oxa + O3a + $T_{amb}$** | **2** | **5** | **extratrees** | **0.86** | **3.08** | **2.36** |
| $O_3$ ~ Oxa + O3a | 2 | 5 | extratrees | 0.82 | 4.16 | 3.21 |
| $O_3$ ~ Oxa + O3a + $RH_{int}$ + $T_{int}$ + ToD + wd + ws | 4 | 5 | extratrees | 0.86 | 3.64 | 2.80 |
| $O_3$ ~ Oxa + O3a + $RH_{int}$ + $T_{int}$ + ToD + wd | 4 | 5 | extratrees | 0.86 | 3.61 | 2.75 |
| $O_3$ ~ Oxa + O3a + $RH_{int}$ + $T_{int}$ + ToD | 2 | 5 | extratrees | 0.85 | 3.50 | 2.63 |
| $O_3$ ~ Oxa + O3a + $RH_{int}$ + $T_{int}$ | 2 | 5 | extratrees | 0.81 | 4.15 | 3.23 |
| $O_3$ ~ Oxa + O3a + $RH_{int}$ | 2 | 5 | extratrees | 0.81 | 4.40 | 3.40 |
| $O_3$ ~ Oxa + O3a + $T_{int}$ | 2 | 5 | extratrees | 0.79 | 4.33 | 3.37 |

For random forest, the most accurate NO2 model was determined to be one that included Oxa, O3a, and ambient RH. The optimal mtry parameter was determined to be 2, with a minimum node size of 5. For predicting O3 the results were similar to those of NO2, except that ambient T replaced ambient RH. For both NO2 and O3 the use of ambient T and RH produced more accurate models. Overall the random forest models performed very similarly to the MLR models, with only slight differences in R2 and RMSE.


### 3.4.5 Step 5: Model validation

For MLR and RF, the R2 and RMSE for each block were saved and plotted (Figure 9a-d). As can be seen, the models using ambient T and RH for both O3 and NO2 remained relatively stable across all blocks. They consistently have a higher R2 and a lower RMSE than the models trained with internal T and RH, for both NO2 and for O3. Conversely, the models trained

with internal T and RH are much more volatile in terms of R2 and RMSE, for both NO2 and O3. In addition, blocks 11, 12, and 13 show a marked decrease in R2 and increase in RMSE across all models with internal T and RH. This trend was true for several models tested at this step, indicating that the internal T and RH were less stable for these blocks. Generally, the differences in RMSE between ambient and internal T and RH were more pronounced for NO2 than for O3. This is true across most blocks and indicates that the final concentrations should be predicted using ambient T and RH data instead of

internal. Tables 6 and 7 show the median R2 and RMSE for all selected models for NO2 and O3, respectively. They reveal

MLR and RF using ambient T and RH are similarly accurate at predicting NO2 and O3. The differences in accuracy are more pronounced for the models using internal T and RH.

Of all predictors included in the RF models, the MOS variable O3a had the highest VI for predicting both O3 and NO2
(Figure 10a-d). The MOS variable Oxa was also of relative importance, usually as the 2nd most important variable, with the exception of the O3 models for which temperature (internal or ambient) was sometimes the 2nd most important variable. Results from these graphs indicate that all variables should remain in the RF models.

**Table 6. Median R² and RMSE across all test blocks of the best MLR and RF models using internal and ambient T and RH for**
**NO2. RMSE and MAE are reported in units of ppb.**

| | $NO_2$ | Median $R^2$ | Median RMSE | Median MAE | |
|---|---|---|---|---|---|
| MLR | $NO_2 \sim \log(Oxa) * O3a * RH_{amb} * (1/T_{amb})$ | 0.82 | 4.35 | 3.54 | *Model 1a* |
| | $NO_2 \sim \log(Oxa) * O3a * RH_{int} * (1/T_{int})$ | 0.67 | 6.12 | 4.10 | *Model 1b* |
| RF | $NO_2 \sim Oxa + O3a + RH_{amb}$ | 0.79 | 4.37 | 3.56 | *Model 2a* |
| | $NO_2 \sim Oxa + O3a + T_{int}$ | 0.72 | 5.29 | 3.89 | *Model 2b* |

**Table 7. Median R² and RMSE across all test blocks of the best MLR and RF models using internal and ambient T and RH for O3.**
**RMSE and MAE are reported in units of ppb.**

| | $O_3$ | Median $R^2$ | Median RMSE | Median MAE | |
|---|---|---|---|---|---|
| MLR | $O_3 \sim \log(Oxa) * O3a * (1/RH_{amb}) * T_{amb}$ | 0.91 | 3.83 | 2.86 | *Model 3a* |
| | $O_3 \sim \log(Oxa) * O3a * (1/RH_{int}) * T_{int}$ | 0.82 | 4.81 | 3.79 | *Model 3b* |
| RF | $O_3 \sim Oxa + O3a + T_{amb}$ | 0.90 | 3.77 | 3.00 | *Model 4a* |
| | $O_3 \sim Oxa + O3a + T_{int} + RH_{int} + ToD$ | 0.86 | 5.20 | 4.10 | *Model 4b* |





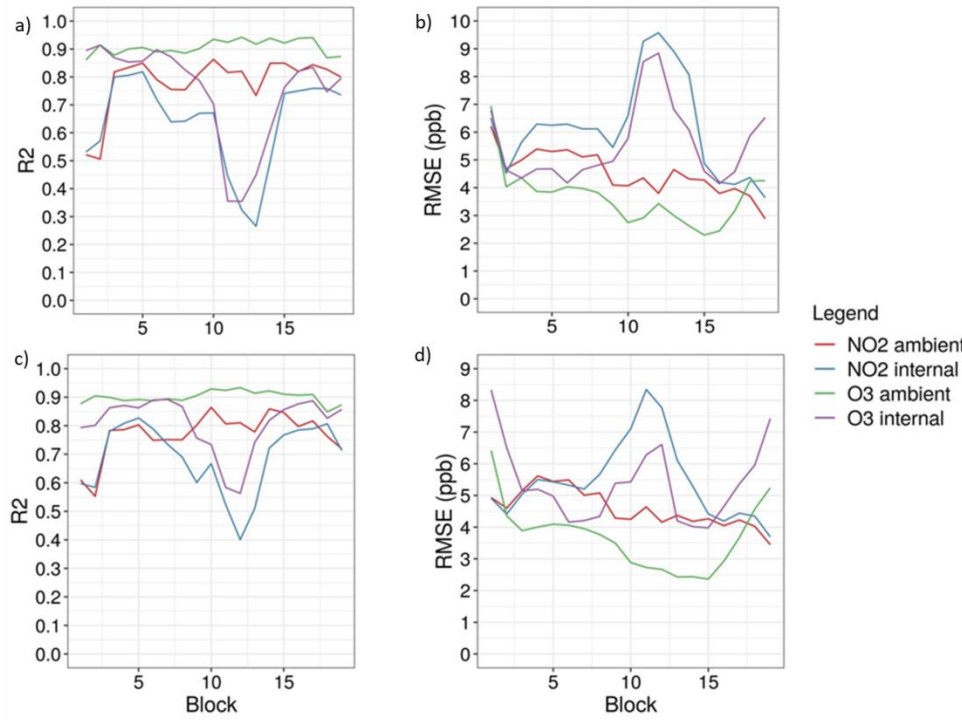

**Figure 9. a) R² and b) RMSE over the 19 test blocks for the MLR models (1a, 1b, 3a, 3b), respectively. c) R² and d) RMSE over the 19 blocks for the RF models (2a, 2b, 4a, 4b), respectively.**

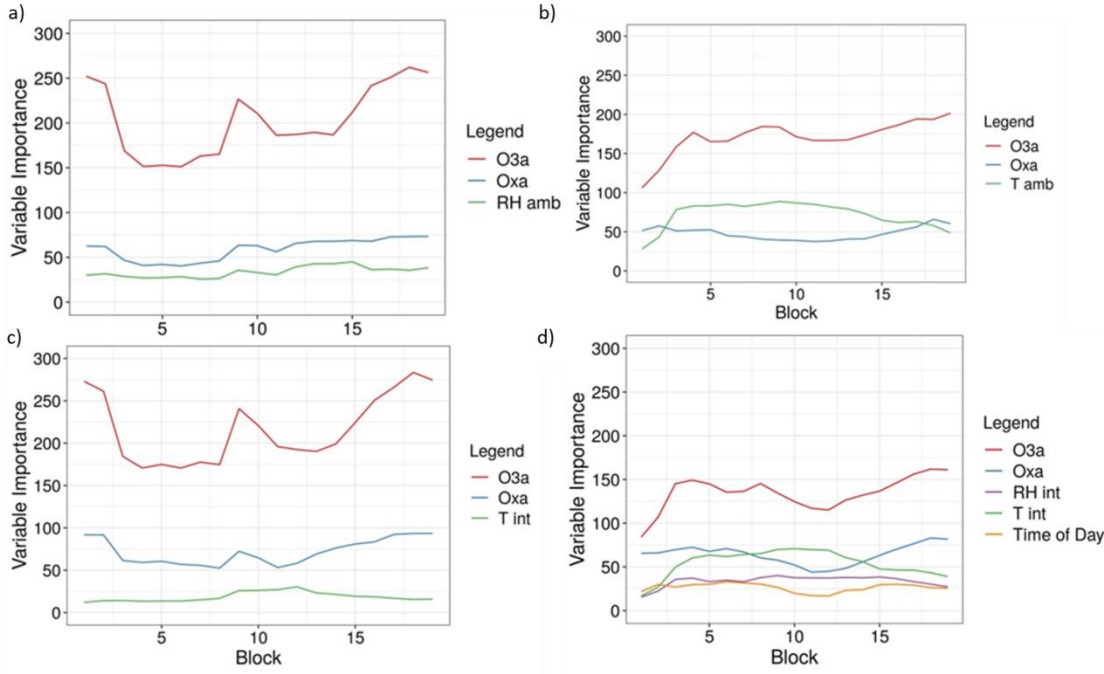

**Figure 10. Variable importance over the 19 test blocks of a) model 2a, b) model 4a, c) model 2b, and d) model 4b.**





### 3.4.6 Step 6: Predicting final concentrations

Final concentrations predicted for NO2 and O3 using the MLR and RF models with both ambient and internal T and RH can be seen in Figure 11. While the results indicated that ambient T and RH should be used, both are included here for further analysis beyond the seven step methodology. For NO2, the MLR models predict a much narrower range of concentrations and occasionally predict negative concentrations (Figure 11a). The RF models tend to predict higher concentrations than

MLR, have a wider range, and don't predict negative concentrations (Figure 11b). For O3, the differences between MLR and RF are less pronounced, with both capturing the diurnal cycle well (Figure 11c-d). In all figures it can be seen that models using ambient T and RH consistently predict higher concentrations than those using internal T and RH. This indicates that there is a difference between predictions using Zephyr internal versus reference temperature and relative humidity sensors.

### 3.4.7 Step 7: Calculating predictive error

As can be seen in Figure 12, smearing the reference data had minimal impact on the predictive accuracy of all models. This indicates that the uncertainty of the reference instruments did not impact the predictive accuracy of the models and can therefore be ignored as a potential interference. Overall predictive error was then calculated as the reference error plus median MAE of each model across all blocks from the model validation step. The T-200 NOx instrument has a measurement

uncertainty of 0.5% of the measurement above 50 ppb or an uncertainty of 0.2 ppb below 50 ppb. For the Tech 2B Ozone Monitor, the uncertainty was the larger between 2% of the measurement or 1 ppb. This can be seen in Figure 13, which depicts the MLR and RF predicted concentrations for Experiment 1 with shaded regions representing the uncertainty. The uncertainty between RF models and MLR models was fairly similar, but was higher for NO2 than for O3. This reflects the findings from Steps 4-6 in which O3 was predicted more accurately than NO2 by both models. The standard error for MLR

models was found to not reflect the realistic accuracy of the predicted concentrations in relation to actual concentrations, as it was found to be very low. The RF models calculated a more appropriate measure of standard error using the infinitesimal jackknife method (Wager et al., 2014), but for consistency with the MLR models, this measure was not used. The accuracy of the final models in predicting on experimental data for which reference concentrations are not available for comparison is then best reflected by combining the uncertainty of the reference instruments with the median MAE of the test blocks during

Step 5 (Tables 6 and 7).

### 3.5 Extra validation step

   To further test the impact of using more representative training datasets, the final models identified in Steps 4 and 5 were trained with each co-location individually as well as with both combined. The predictive accuracy of these separate models

was then compared using the experimental dataset for which reference NO2 and O3 measurements were available, as Zephyr s71 was co-located throughout the experiment. Additionally, these datasets were also tested with data points flagged in Step 3 removed to understand further influences on model accuracy. This extra validation allowed for better evaluation of the





performance in predicting experimental concentrations of the MLR and RF models selected with the seven-step method. This is, however, atypical for field studies, as these sensor systems are intended to be deployed independently of reference
instrumentation.

Table 8 shows the results of training these various models for NO2. The most accurate model at predicting experimental concentrations was the RF model using internal T and trained with data only from co-location 2. The same model trained with all available co-location data was slightly more inaccurate. Co-location 3 was the least accurate of the training subsets,
reiterating findings from Step 1. For the MLR models, this dip in accuracy when using exclusively co-location 3 as the training set was most pronounced, as can be seen in Table 8 and Figures 14g-h. When filtering out flagged data points, most NO2 models improved slightly in predictive accuracy. This was most pronounced for those using co-location 3 as a training set, which improved substantially in terms of R2.

For O3, the most accurate model was the RF model using internal T and RH and trained exclusively using data from co-location 2, though the MLR internal model for the same co-location was of comparable accuracy. The RMSE for this model was substantially lower than the one trained using ambient T and RH. With the MLR models, this difference in predictive accuracy between models trained with internal and ambient T and RH was much greater, again favoring the internal models. Co-location 3 was highly inaccurate at predicting experimental data, further reiterating findings from Step 1 that indicated
the unsuitability of this co-location for use in predicting final concentrations. Figures 15e-f clearly depict the boundaries for predictions with RF models when the training data are unsuitable, as is the case with co-location 3. Filtering out the points flagged in Step 3 did not improve the predictive accuracy of models trained exclusively with co-location 2, but it substantially improved those trained with co-location 3, especially those using internal T and RH.




**Figure 11. Time series plots and boxplots for Experiment 1 of a) predicted NO2 concentrations using the MLR model, b) predicted NO2 concentrations using the RF model, c) predicted O3 concentrations using the MLR model, d) predicted O3 concentrations using the RF model. 'Ambient' and 'internal' refer to the use of ambient or internal T and RH data in each model.**





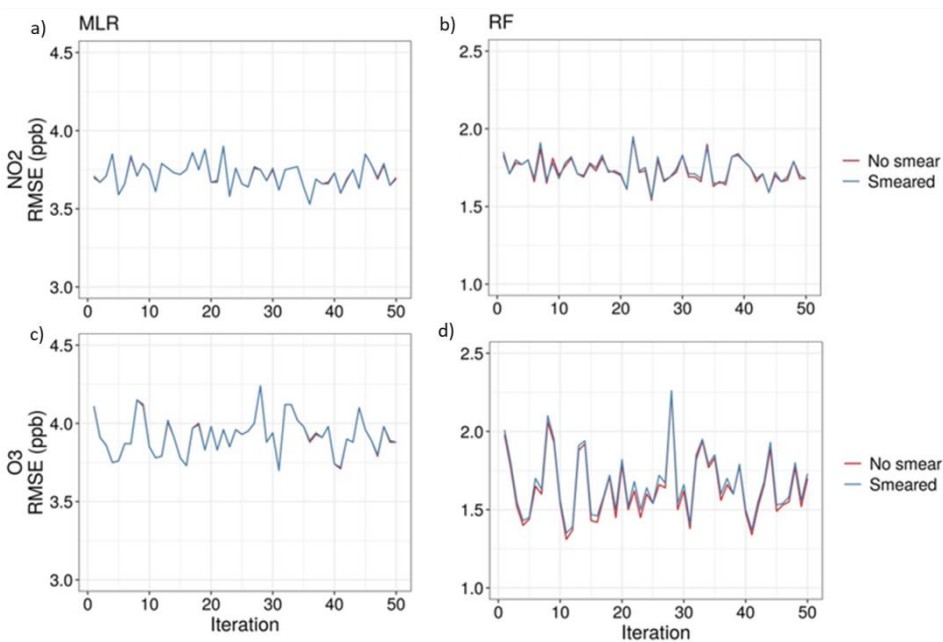

**Figure 12. RMSE of models trained using smeared reference measurements versus actual reference measurements for a) NO2 with MLR, b) NO2 with RF, c) O3 with MLR, and d) O3 with RF.**

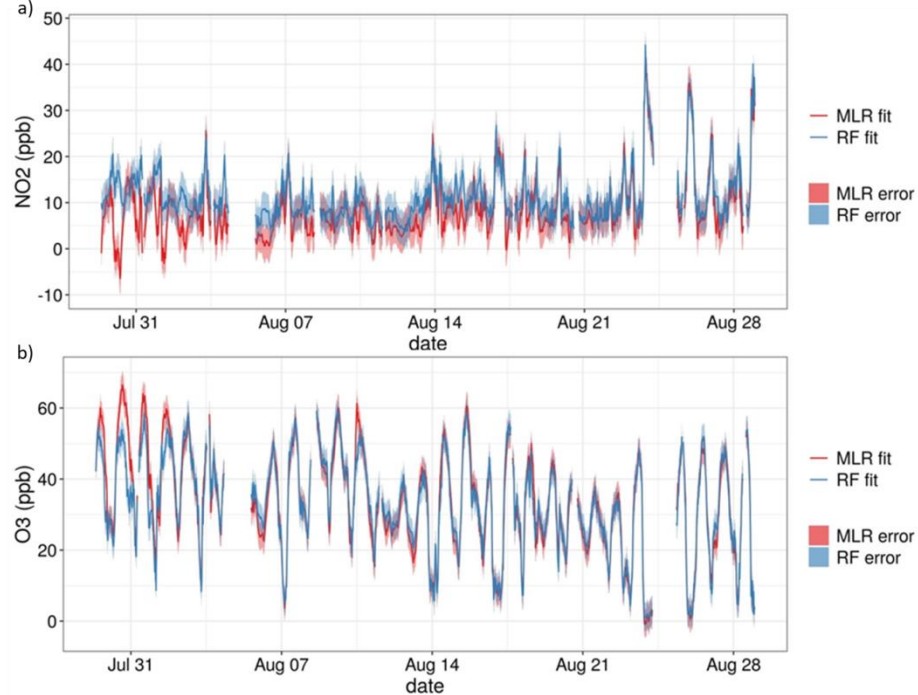

**Figure 13. Time series plots of both MLR and RF predictions for Experiment 1 including the measurement uncertainty as shaded regions for a) NO2 and b) O3. Data were averaged to 30 minute resolution.**






**Table 8. Results of RF and MLR models for NO2 trained with co-location 2, co-location 3, or a combination of both when tested on the comparison experimental dataset. In the lower half of the table, the models are trained with the same datasets but are tested on the experimental dataset with data points outside the ranges of each training dataset filtered out.**


| | $NO_2$ | Co-location 2 | | Co-location 3 | | Both co-locations | |
|---|---|---|---|---|---|---|---|
| | | $R^2$ | RMSE | $R^2$ | RMSE | $R^2$ | RMSE |
| MLR | $NO_2 \sim \log(Oxa) * O3a * RH_{amb} *(1/T_{amb})$ | 0.66 | 5.49 | 0.22 | 12.08 | 0.61 | 5.66 |
| | $NO_2 \sim \log(Oxa) * O3a * RH_{int} * (1/T_{int})$ | 0.66 | 6.41 | 0.57 | 10.99 | 0.67 | 5.55 |
| RF | $NO_2 \sim Oxa + O3a + RH_{amb}$ | 0.70 | 5.07 | 0.48 | 6.04 | 0.59 | 5.43 |
| | $NO_2 \sim Oxa + O3a + T_{int}$ | **0.73** | **4.42** | 0.61 | 5.57 | 0.68 | 4.87 |
| | | | | | | | |
| | $NO_2$ – filtered | | | | | | |
| MLR | $NO_2 \sim \log(Oxa) * O3a * RH_{amb} *(1/T_{amb})$ | 0.63 | 5.53 | 0.45 | 12.88 | 0.62 | 5.62 |
| | $NO_2 \sim \log(Oxa) * O3a * RH_{int} * (1/T_{int})$ | 0.63 | 6.49 | 0.64 | 11.17 | 0.69 | 5.54 |
| RF | $NO_2 \sim Oxa + O3a + RH_{amb}$ | 0.68 | 4.88 | 0.65 | 5.93 | 0.59 | 5.38 |
| | $NO_2 \sim Oxa + O3a + T_{int}$ | 0.71 | 4.38 | 0.65 | 6.01 | 0.68 | 4.85 |

**Table 9. Results of RF and MLR models for O3 trained with co-location 2, co-location 3, or a combination of both when tested on the comparison experimental dataset. In the lower half of the table, the models are trained with the same datasets but are tested on the experimental dataset with data points outside the ranges of each training dataset filtered out.**

| | $O_3$ | Co-location 2 | | Co-location 3 | | Both co-locations | |
|---|---|---|---|---|---|---|---|
| | | $R^2$ | RMSE | $R^2$ | RMSE | $R^2$ | RMSE |
| MLR | $O_3 \sim \log(Oxa) * O3a * (1/RH_{amb}) * T_{amb}$ | 0.86 | 7.00 | 0.86 | 5.12 | 0.88 | 6.06 |
| | $O_3 \sim \log(Oxa) * O3a * (1/RH_{int}) * T_{int}$ | 0.94 | 3.37 | 0.16 | 17.20 | 0.91 | 3.94 |
| RF | $O_3 \sim Oxa + O3a + T_{amb}$ | 0.91 | 5.16 | 0.74 | 7.65 | 0.91 | 5.13 |
| | $O_3 \sim Oxa + O3a + T_{int} + RH_{int} + ToD$ | **0.94** | **3.30** | 0.67 | 9.91 | 0.92 | 3.82 |
| | | | | | | | |
| | $O_3$ – filtered | | | | | | |
| MLR | $O_3 \sim \log(Oxa) * O3a * (1/RH_{amb}) * T_{amb}$ | 0.85 | 6.78 | 0.85 | 4.13 | 0.87 | 5.97 |
| | $O_3 \sim \log(Oxa) * O3a * (1/RH_{int}) * T_{int}$ | 0.93 | 3.33 | 0.52 | 9.24 | 0.91 | 3.90 |
| RF | $O_3 \sim Oxa + O3a + T_{amb}$ | 0.91 | 5.18 | 0.77 | 5.13 | 0.90 | 5.15 |
| | $O_3 \sim Oxa + O3a + T_{int} + RH_{int} + ToD$ | 0.93 | 3.30 | 0.65 | 7.53 | 0.91 | 3.82 |





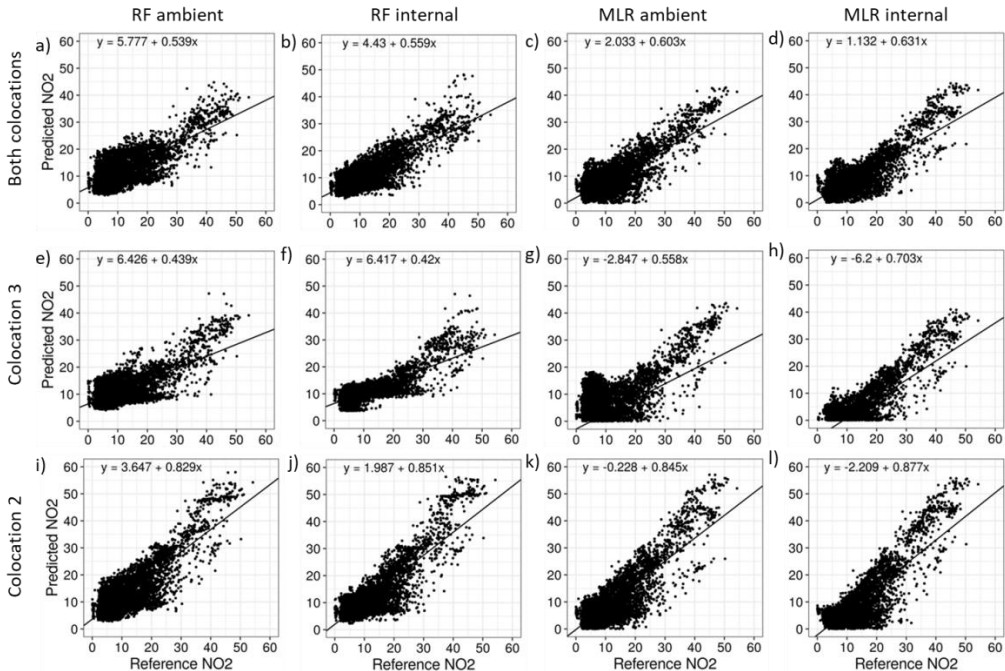

**Figure 14: Scatter plots of predicted NO2 versus reference NO2 concentrations for the experimental data using MLR and RF models trained with co-location 2 (i-l), co-location 3 (e-h), and both combined (a-d). All concentrations are reported in ppb.**

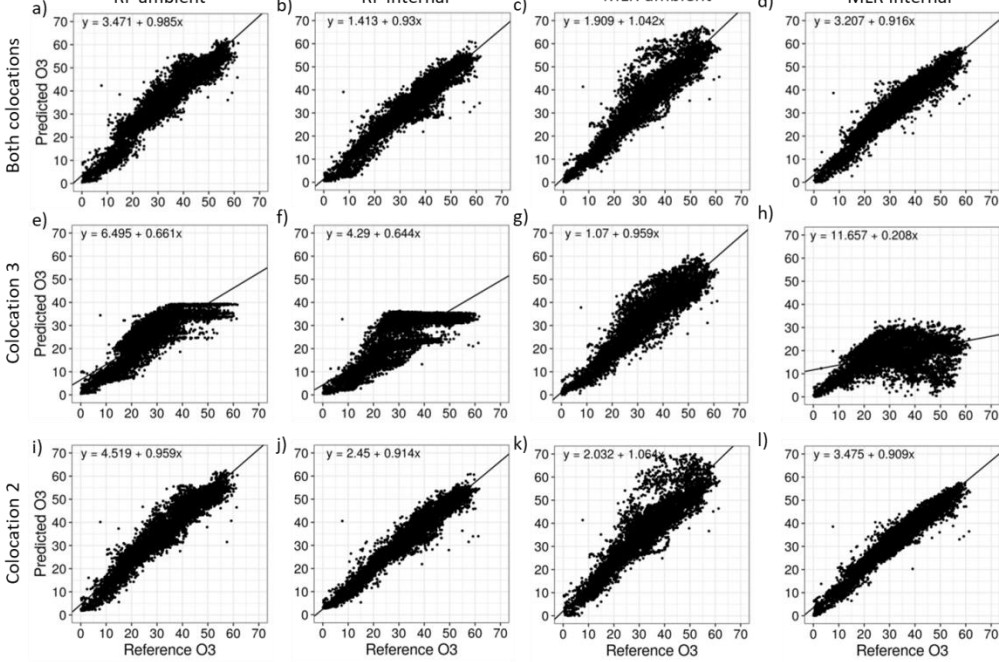

**Figure 15. Scatter plots of predicted O3 versus reference O3 concentrations for the experimental data using MLR and RF models trained with co-location 2 (i-l), co-location 3 (e-h), and both combined (a-d). All concentrations are reported in ppb.**




## 4. Discussion

The results of this study have several implications for the field of low-cost sensors. In line with other research, this study found that MLR and RF were similarly accurate in predicting experimental concentrations of NO2 and O3 (Karagulian et al., 2019), though the differences in accuracy between MLR and RF were more pronounced for O3 than for NO2. In fact, it was

found that RF was the better predictor of both O3 and NO2 concentrations when evaluated with the longer experimental data set, albeit only slightly. This contrasts with findings from the model selection and validation process, as the MLR models were consistently more accurate at predicting on subsets of the co-location data. What this indicates is that models found to be more accurate during "calibration" may have differing model performance when assessed with a "comparison" dataset, in this case the experimental dataset that was co-located throughout for one sensor. This is a result that has been found

previously, where the R2 is lower for comparison datasets than for calibration (Karagulian et al., 2019). If RF, MLR, or other ML techniques are selected for their accuracy when predicting on calibration data and are not tested on comparison data, it may well be that the performance does not hold for new experimental data. Given the similarity between RF and MLR in predicting NO2 and O3 found in this study, as well as in the literature, either method can be used. However, as MLR is simpler to implement than most ML techniques, has fewer parameters that need to be optimized, and the model calculations

are well understood, unlike the black-box calculations of RF and most ML techniques, this is should be the preferred option for those who enjoy greater model transparency and control.

Further important to the proper evaluation of model accuracy is the reporting of multiple metrics such as RMSE and MAE, in addition to $R^2$. It is quite clear from Tables 8 and 9 that $R^2$ is not the best metric with which to measure predictive

accuracy of calibration models. Models trained with co-location 3 exclusively to predict O3, for example, had an $R^2$ greater than 0.70, which is considered to be acceptable agreement. Those same models, however, had an RMSE of >7 ppb, which is much more inaccurate than an $R^2$ of 0.70 alone would reveal. As another example, the same models trained exclusively with co-location 2 for O3 (Table 9) had an $R^2$ between 0.86 and 0.94 (very good performance), but had a wide range of RMSE between 3.30-7.00 ppb. It is therefore crucial that multiple performance metrics are used to evaluate calibration models

before final decisions are made on their suitability. At a minimum, $R^2$ and RMSE should be reported.

Multicollinearity is an issue common not only to MLR, but also to small sensor systems, which often have multiple LCSs measuring the same or similar species with heavily auto-correlated data. While uncommonly addressed in the literature, except for a few studies mentioning its influence on MLR models (Bigi et al., 2018; Hagan et al., 2018; Masiol et al., 2018),

the solution, as presented in Steps 4 and 5, is relatively straightforward. To ensure that the predictor variables included in the final model are, in fact, explanatory, the model should be repeatedly validated using bootstrapped samples. To deal with autocorrelation, this validation should be done using continuous blocks and not with random sampling. Including these steps in the model-building process is simple and should be considered best practice.



Further underlining the importance of repeated validation is the variation in results when using ambient or internal T and RH. While the inclusion of ambient meteorological data led to more accurate models during calibration, this did not hold for the comparison dataset. Instead, for the prediction of both NO2 and O3, it was internal T and RH data that led to more accurate prediction. This indicates that for the prediction of NO2 and O3 concentrations with EarthSense Zephyrs, not only are the internal T and RH sensors acceptable for use in predictive models, but they are likely more representative of normal 620 operating conditions. Given that the MOS sensors radiate large amounts of heat, the conditions inside the Zepyhrs are significantly different than, but linearly related to, ambient conditions. As such, the internal T and RH sensors likely better represent the exact environmental conditions under which species are adsorbing to the MOS sensors.

The final results also reveal the value of pre-processing the data in Steps 1-3. It became clear by looking at the distribution of 625 the co-location datasets in Step 1 that co-location 3 might be unsuitable for use in predicting the experimental concentrations. These data were then flagged in Step 3. While the models trained exclusively with co-location 3 were substantially less accurate than those using data from co-location 2, their accuracy increased when flagged experimental data points outside the range were removed. In essence, the 3rd co-location was useful for predicting on experimental data within its range of conditions, but very inaccurate for those outside of that range. Co-location 2, on the other hand, was identified as 630 being well-suited for prediction in Step 1 and received few flagged points in Step 3. Final results indicate that MLR and RF models trained with co-location 2 perform better than those trained with co-location 3, for both NO2 and O3. Combining the two co-locations did not improve the predictive accuracy for NO2 or O3, when compared with the more-suitable 2nd co-location (Tables 8 and 9). As such, training calibration models with co-location 2 exclusively would have been correctly justified using evidence from Step 1. What is evident from this analysis is that ensuring quality of training data used in 635 calibration is crucial to accurate prediction. Incorporating quality control into the calibration methodology is therefore an important best practice.

Finally, LCS data should be reported with associated error values. While we discussed RMSE in the context of model fit and validation, as well as a method for evaluating whether reference instrument accuracy affects the model output, error values 640 should be reported not just in the assessment of the LCSs themselves, but also with some form of representative error associated to the reported concentration data. Our recommendation is to combine the uncertainty of the reference instruments with the median MAE across blocks from the model validation step. As can be seen in Tables 8 and 9, the RMSE of predictions tested with the comparison experimental dataset are quite similar to the median RMSE values in Tables 6 and 7. This indicates that using median error from the model validation step is quite representative of the LCS uncertainty. 645 However, over longer measurement campaigns, this should be repeatedly tested and validated with additional co-location training sets, so as to account for sensor drift, deteriorating functionality, and varying meteorological conditions.



## 5. Conclusions

While many details of this methodology are already well-known, they are often overlooked or go unreported in published
literature. In most cases not all aspects are included. As a result, many studies assessing pairwise calibration methodologies
for low-cost sensors cannot be compared. In the absence of calibration standards for these technologies in a field that
continues to diversify and grow, researchers must start to consolidate around an agreed-upon set of best practices. This study
has highlighted several of them. First, details on model selection, validation, and tuning must be reported if researchers are to
be able to effectively compare results across studies. If models are not rigorously tested for suitability using standardized
methods, especially with "black-box" machine-learning techniques, then their comparison will remain challenging at best.
Second, models should be validated not only on the calibration dataset, but also on a separate comparison dataset, if possible.
All validation should be done using R² and RMSE, at a minimum. This will provide greater insight into the suitability of
selected models for prediction on experimental data as well as better comparability across studies. Third, pre-processing the
data, including visual inspection, outlier removal, and data-flagging are an integral part of an effective calibration
methodology. Understanding the quality and distribution of available data is important to identifying problems and solutions
encountered during calibration.

Last, it is clear that a standardized methodology for the calibration of low-cost sensors is needed if they are to be
incorporated into air quality monitoring programs and contribute new insights to the field of atmospheric chemistry. This
seven-step methodology seeks to fill a gap in the literature up until now left largely unreported. In addition, this
methodology, complete with relevant R code, is the first to be completely transparent and open-access. This is a valuable
contribution to a young, but rapidly growing body of literature surrounding low-cost sensors. With this work, we hope to
begin pulling back the curtains on the black box of sensor calibration.

**Code availability**

All relevant code for this study can be found in this open-access Zenodo repository: https://doi.org/10.5281/zenodo.4317521

**Data availability**

All relevant data for this study can be found in this open-access Zenodo repository: https://doi.org/10.5281/zenodo.4309853

**Acknowledgements**

The authors would like to thank Mark Lawrence for his support of the research work, Martin Schultz, Clara Betancourt, and
Phil Peterson for their discussions on the subject, Achim Holtmann for his support of the meteorological data collection, and
Roland Leigh, Jordan White and the EarthSense team for their collaboration and support. The research of EvS, SS, AC, GV
is supported by IASS Potsdam, with financial support provided by the Federal Ministry of Education and Research of
Germany (BMBF) and the Ministry for Science, Research and Culture of the State of Brandenburg (MWFK). Much of the



research was carried out as part of the research program 'Urban Climate Under Change' [UC]2, contributing to 'Research for Sustainable Development' (FONA; www.fona.de), within the joint-research project 'Three-dimensional observation of atmospheric processes in cities (3DO)', funded by German Federal Ministry of Education and Research (BMBF) under grant
01LP1602E.

**Author Contributions**

SS developed the method, conducted the analysis, and wrote the paper. ST supported the method development and contributed to revisions and editing. GV conducted the measurements, contributed to the concept development for the
measurement campaigns, and contributed to editing. AC contributed to the method development and data analysis, as well as writing and editing. IL, FM collected and provided processed meteorological data in the context of the measurement campaigns, as well as writing and editing. DK and RW did the comparison of the reference instruments. EvS developed the concept and contributed to the measurements, method development, writing, editing and overall coordination. All authors gave final approval for publication.


**Competing interests**

The authors declare that they have no conflict of interest.

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
