# Peer review of "Unraveling a black box: An open-source methodology for the field calibration of small air quality sensors"

_Atmospheric Measurement Techniques, 2020_

## Author Comment (AC1)

**Reviewer #1**

This manuscript provides a seven-step methodology for the calibration and quality assurance of low-cost air quality sensors. Thanks to the generalised nature of this method, it can be applied to a wide range of sensors and potentially be used as a standard calibration procedure. The data processing script was made publicly available which maximises the applicability of this method and the impact of this research.

The authors have pointed out current challenges in the use of low-cost sensors including the lack (or incomparability) of calibration procedures in many low-cost sensor application studies. They stress the need of a reliable and reproducible data calibration and post-processing method. This manuscript is an important step towards this aim and, therefore, a valuable contribution to the literature in this field as it has the potential to improve the data quality in future applications of low-cost sensors. The manuscript is well structured and clearly written.

My main suggestions to further improve the scientific quality of the manuscript are:

- Discuss the limitations of this method in more detail (Point 1)
- Add physical explanations of the found observations (Point 5)

*We thank the reviewer for their positive comments regarding this manuscript and their recognition of the need for such a standardized methodology in the calibration of low-cost sensors. Their comments are very helpful and have improved the quality of this manuscript. What follows are point by point responses to their main comments. All technical comments were accepted and changed in the manuscript, unless otherwise specified.*

**Specific comments**

**1. Please discuss the limitations of your calibration method in more detail (Point 2.1)**

- **Application range of calibrated sensors (indoor vs outdoor vs mobile)**

  You stressed the importance of calibrating the sensors under conditions that are similar to those under which they will be (or have been) operated during the experimental application. This needs to be considered when defining the application range of the sensors.

  Thanks to their silent operating conditions and small size, low-cost sensors are suited for indoor as well as mobile applications (e.g. wearable sensors for personal exposure assessment). However, if the calibration is conducted outdoors, the sensors might not be suited for such applications as the environmental conditions may differ significantly in these environments. Furthermore, mobile deployments would require further data cleaning and validation steps as rapidly changing environments may have an impact on the sensor performance (e.g. Alphasense Ltd., 2013).

*We thank the reviewer for this comment. Some text has been added to this section to clarify the study's focus on stationary field deployment and to highlight that different calibration considerations should be used depending on the environment in which the sensors will be deployed.*

*The added text reads, "In this study, we focus primarily on stationary field deployment of low-cost sensor systems. There are, however, other forms of deployment, including indoor and mobile, for which these criteria also apply. It is important to mention that there may be other considerations required in such alternative forms of deployment, e.g. more scrutinous data cleaning in mobile deployments due to impacts of rapidly changing environments on sensor performance."*

- **Sensor systems**

  As you have pointed out, low-cost sensors are often temperature and RH dependent as well as cross-sensitive to other pollutants. Therefore, it should be recommended to apply the presented calibration method to sensor systems (with additional sensors for T, RH and cross-sensitive gases) rather than individual sensors.

*We thank the reviewer for this comment. In the manuscript we have tried to be coherent in describing that this methodology is intended for use on small sensor systems, which often include individual sensors for temperature, relative humidity, and other cross-sensitivities. We have gone through the manuscript to ensure that this is consistently presented. Please see the track changes version of the manuscript for these changes.*

- **Data cleaning (Point 2.2.2)**

  In this step, point outliers are removed based on the assumption of a slowly changing airfield where peak exposures over a few seconds do not occur. However, such short-term (< 10 sec) emissions may occur in certain settings (e.g. traffic emissions of nearby passing vehicles, cigarette emissions of passengers etc.). One advantage of the high spatial and temporal resolution of low-cost sensors is that such peak exposures may be captured. The proposed method, however, excludes such events. Please include this argument when defining the application range of the sensors (Point 2.1).

*We thank the reviewer for this comment. Indeed, this is a significant challenge in data cleaning and unfortunately requires, in some cases, subjective assessment for an accurate determination to be reached. While it is possible that this data cleaning method removes some non-outlier measurements during peak emissions, it is equally possible that such events are indeed outliers due to technical sensor error. For this reason, we recommend that identified outliers be graphically compared with neighboring points to determine if their removal is justified.*

*With the optimized moving window and threshold identified in this study, for sensor system s71, a total of 58 outliers were detected from >500,000 data points. In this case individual assessment of each point's 'outlierness' was practical, but there may be cases where this is impractical. In such cases, we recommend a random subset of outliers be graphically assessed to determine the extent to which the data cleaning function is removing actual outliers. This is imperfect, but it is unlikely that a data cleaning method exists which can perfectly separate outliers from peak events. It is with such peak events that other tested methods such as the AutoRegressive Integrated Moving Average performed particularly poorly, identifying most peak events as outliers. If such events are expected due to the deployment environment, particular care in the evaluation of potential outliers should address this.*

*Clarification text has been added to section 2.2.2 regarding this and now reads, "The points flagged as outliers with this method were then graphically assessed against neighboring datapoints to prevent inadvertent removal of peak emission events. In other cases where assessing all outliers is impractical, it is recommended to do so with a random subset of outliers. Furthermore, if substantial short-term events are expected due to the deployment environment, such as during mobile measurements, a more thorough check of potential outliers should be done."*

**2. Line 115:** You state that, while demonstrated here with MOS, the proposed calibration method can equally be applied to electrochemical sensors. To strengthen this argument, please add a brief physical explanation, a reference, or experimental proof.

*We thank the reviewer for their comment. A brief physical explanation has been added in the text. While different in their design, both MOS and EC sensors produce a measure of voltage/resistance which varies in response to changing concentrations of gas-phase species, and hence can be calibrated using the same methodology. In more recent published work, we have successfully applied this methodology for the calibration of EC sensors, see Schmitz et al., (2021).*

*The added text reads "Furthermore, while it was applied here to sensor systems containing metal oxide LCS, this methodology is also equally as applicable to electrochemical LCS or photoionization detectors (PID), as these produce a similar measure of voltage that varies in response to changing concentrations of gas-phase species and have similar cross-sensitivities to temperature and relative humidity. It is not directly applicable for optical particle counters (OPC) for the measurement of particulate matter, as the transformation of the raw data into concentrations during calibration functions differently, though some of the principles discussed here are still relevant. For an application of this methodology to EC sensors, please see Schmitz et al., (2021)."*

**3. Line 221, line 240:** Please explain how you have determined the splitting ratio between training and validation period. How much differ you results when using other ratios?

*We thank the reviewer for their comment. In this case a standard splitting ratio of 75:25 for training and validation test sets was used, as this is common practice in model building. However, we have conducted a robustness cross-check with various splitting ratios and found that our results did not differ substantially when using other ratios. The training to validation splitting ratios tested were 90:10, 80:20, 75:25, 70:30, 60:40, and 50:50. For MLR, the median $R^2$ across all blocks in Step 5 for $NO_2$ models ranged between $0.78 - 0.83$ for ambient T/RH and between 0.59-0.74 for internal T/RH. For $O_3$ models, the median $R^2$ ranged between $0.90 - 0.93$ and $0.60 - 0.85$ for ambient and internal T/RH, respectively. For RF, the median $R^2$ across all blocks for $NO_2$ ranged between $0.71 - 0.77$ and $0.61 - 0.74$ for ambient and internal T/RH, respectively. For $O_3$ the median $R^2$ ranged between $0.89 - 0.93$ and $0.70 - 0.89$ for ambient and internal T/RH, respectively. Given these results, we feel that using a splitting ratio of 75:25 is justified, as the results do not differ significantly based on this choice.*

*Text has been added in section 2.2.5 that reads, "A robustness cross-check with various splitting ratios was conducted and found that changing the splitting ratio did not significantly impact the results."*

**4. Table 6:** Please explain why you are using the medians and not the means of your statistical parameters. (whereas in Line 221 you were speaking about the average RMSE)

*We thank the reviewer for pointing out this discrepancy. The median was chosen instead of the mean as it is less susceptible to the influence of extreme values. The discrepancy is due to the*

*nature of the R function 'train()' from the 'caret' package used in Step 4, which provides only the mean RMSE during operation. In step 5 the calculations were done manually and thus the median was preferred. To correct this, the mean RMSE, MAE, and $R^2$ from Step 4 were replaced with manually calculated medians. This change is reflected in the text and in tables 2-5.*

**5.** While the manuscript nicely discusses the implications of a finding, it sometimes does not offer **physical explanations** for them:

> • **Line 245:** "If the graphs showed instability across the various folds, Step 4 was repeated and a new model was selected for validation"
>
> What causes this instability and how can you ensure that the model stays stable under field conditions?

*We thank the reviewer for this comment. A sentence for clarification has been added to the text. We use instability to refer to major changes in $R^2$ and RMSE between folds in the model validation process. This is likely caused by differences in field conditions between the training and test folds. The best way to ensure the model remains stable under field conditions is with repeated co-location over longer time periods, in coordination with meteorological changes due to seasonality. The more training data available for calibration, the better the chances that the final model will be stable under field conditions.*

*The added text now reads, "In this case, instability refers to major differences in $R^2$ and RMSE between folds likely caused by differing field conditions among the training and test folds. If this is seen, it indicates that the model may be too sensitive to changes in field conditions."*

> • **Section 3.4.4 (model selection):** Different relationships between the input variables were found for different models, e.g. an inverse temperature dependence for $NO_2$ was found for the best fitting MLR but no temperature dependence was found in the case of the best fitting RF. How can you explain this and what type of physical relationship (e.g. temperature dependence) would you expect?

*We thank the reviewer for this comment. A dependence on temperature was expected for the $NO_2$ models and was therefore included during the initial model selection process for both the MLR and RF models. However, the nature of this physical relationship was not clear, as the sensor specifications indicated that expected temperatures during field deployment would not impact the functioning of the MOS sensors. Rather, the dependence on temperature was expected due to the impact that temperature, as a proxy for insolation, has on daytime chemistry. An inverse relationship in this sense makes sense, as $NO_2$ is photolyzed in VOC-sensitive environments to produce $O_3$, which is normally the case in urban environments such as Berlin.*

*Following the update of the tables in Step 4 to reflect the median RMSE/MAE/$R^2$ instead of the mean for each model in response to the reviewer's previous comment, the best RF model for $NO_2$ was found to include T, which was previously not the case. All subsequent tables and graphs throughout the example were updated to reflect this change in the $NO_2$ RF model. Therefore, the reviewer's comment is partially answered, as there is in fact a temperature dependence in the RF model.*

*In the case of MLR, the final temperature dependence was determined to require an inverse transformation, whereas for RF, the relationships are equal in nature, as inverse, logarithmic, etc.*

*transformations do not affect the outcome of the RF model. This is principally due to different calculations that occur within the mechanisms of each model. For an RF model, this involves randomly choosing a variable by which to split the decision tree. This occurs at each node until no more splits are possible or the data are collected into final bins containing 5 data points. Therefore, any physical transformation of the data will not lead to a change in the calculations that occur in an RF model.*

*Text has been added to section 3.4.4, which now reads, "This is in line with what would be expected in urban environments, as T can be seen as a proxy for insolation, which causes the photolysis of $NO_2$."*

- The model performance was found to be higher when using the ambient environmental conditions (T and RH) as parameters (e.g**. Tables 6 and 7**). However, you pointed out in the discussion (**Line 619**) that the internal conditions are more representative for the operating conditions of the sensor. What are possible explanations for this observation?

*We thank the reviewer for this comment. This is indeed an interesting finding that is challenging to explain. The ambient T and RH would be expected to be more accurate models, as they are better representative of the conditions under which chemical processes occur that produce the concentrations of $NO_2$ and $O_3$ measured by the reference instruments. In this regard it makes sense that the model performance during validation was better with models trained using ambient T and RH than with internal. However, since the actual chemical reactions being measured are those that are taking place on the surface of the MOS, it seems that the internal T and RH better represent the conditions of the chemistry inside of the sensor system. The signal produced from the internal T and RH sensors is then used alongside the MOS signal in the models as markers for the chemistry that is occurring inside the sensor system at the time the same parcel of air reaches the sensor system and the reference instrument. However, that there are equally valid explanations for both outcomes warrants a closer investigation into these results. We feel that this would require much more detailed inspection of model predictions and would be outside the scope of this paper, which intends primarily to present and explain a methodology for the calibration of LCS. Future work will take a closer look at these results to determine why this occurs.*

*A sentence has been added to the discussion in line with this comment and another from Reviewer #2 that reads, "However, given that models using ambient data were more accurate during the validation step and significant differences between predictions of models trained with internal vs ambient T and RH were identified, these results require closer inspection, which should be the subject of future research."*

**6. Line 292:** Please specify "decent" and "good" agreement (e.g. with mean R2 & RMSE)

*Done. $R^2$ of these intercomparisons have been added to the text in addition to the reference to the supplemental information.*

**7. Line 327:** You deployed (at least) two low-cost sensors. Have you quantified the agreement between the two sensors? If so, add a small sentence here as it may be a strong argument why it is sufficient to only look at the data of one representative sensor. Perhaps summarise the performance of the second sensor briefly in the main text. How can you explain the non-linear response of sensor s72 (Figure S8)?

*We thank the reviewer for their comment. For this study, we use the two low-cost sensors primarily as examples of how to use the seven-step methodology and did not consider their intercomparison*

*as we felt it might distract from the main focus of the work. However, we have added graphs depicting the agreement between standardized raw LCS data of s71 and s72 during the 2 co-locations of the winter measurement campaign into the supplemental information (Figures S4 and S5). The Oxa and O3a MOS sensors of each sensor system are linearly related, but due to differences in sensor sensitivity, have different baselines. In the summer campaign, the relationship between the O3a sensors of s71 and s72 during co-location 2 is non-linear but returns to linear agreement in co-location 3 and in the winter campaign.*

*A reference to the added graphs in SI and a brief discussion of this point in the text was added [section 3.3] and reads, "To compare sensor performance between s71 and s72, an intercomparison of available co-location raw data was conducted for the oxidizing MOS (Oxa) and ozone MOS (O3a). During all co-locations, the sensors had a linear relationship and an $R^2$ > 0.95 (Figures S4 and S5). In only one instance was this not the case (co-location 2, O3a), where the $R^2$ was 0.59 and a deviation from linearity was detected. This relationship was linear in all other co-locations.".*

**8. Figure 8 (optional):** Adding histograms showing the overlap between colocation and experiment would make the Figure easier to comprehend and help to understand the flagging procedure.

*We thank the reviewer for this comment. We have decided not to include extra figures to the manuscript, as there are already very many. Instead, since the violin plots in Figure 4 would help understand the overlap between co-location and experimental data, we have added text that compares Figure 8 to Figure 4.*

*The added text in section 3.4.3 reads, "This shows the utility of comparing the results of Step 1 with the flags generated in Step 3."*

**9. Line 596:** Replace "for those who enjoy" with "to achieve"

*Done.*

**Technical comments**

**10.** Please use **subscripts** for $NO_2$ and $O_3$ and superscripts for $R^2$ throughout the document.

*Done.*

**11. Lines 93 and 96:** What means SVM? Do you mean SVR (support vector regression)?

*Yes, this was a mix-up between Support Vector Machines and Support Vector Regression. SVM has been changed to SVR to match earlier text in this section.*

**12. Line 149:** Delete "for use in statistical calibration" (the general quality of the final data is likely to be higher)

*Done.*

**13. Line 154 (Style, optional):** Replace "What follows in this section is a" with "This section provides a"

*Done.*

**14. Line 196:** How do you define the range of the colocation data? As the range between the minimum and maximum observations? (Or percentiles?)

*Yes, the range between minimum and maximum observations is meant here. This has been added to the text.*

**15. Line 219:** Please provide references for AIC and VI

*Done.*

**16. Line 263 (optional):** Perhaps add a sentence or reference explaining the term "smearing" as the audience might not be familiar with this practice.

*Done.*

**17. Line 295:** "more information 295 in section 3.2" – this is section 3.2

*This has been changed to "section 3.3", as is correct.*

**18. Table 1:** Is it correct that the sensor models for the reducing and the oxidising gases are identical? (SGX Sensortech MICS- 4514)

*Yes, this is correct. This sensor detects both reducing and oxidizing species.*

**19. Figure 2 (optional):** Adding a timeline with (rough) dates would help to comprehend the paragraph above quicker.

*Figure 2 has been updated and dates have been added to the timeline.*

**20. Figures 4 c, d; 6; 7 a, b; 10 etc:** Make sure that all axes have units (even if only arbitrary units).

*Done.*

**21. Figures 14 and 15 (optional):** Although you have already mentioned them in Tables 8 and 9, add the $R^2$ and RMSE values to the graphs to provide a comprehensive overview.

*Done.*

**22. Line 503:** "the reference instruments did not impact the predictive accuracy of the models and can therefore [in this case] be ignored as a potential interference" – can this be generalised for all sensors? If not, add "in this case"

*Done.*

**23. Line 508:** "The uncertainty between RF models and MLR models was fairly similar" - replace "between" with "of"

*Done.*

**Reference**

Alphasense Ltd. (2013). Alphasense Application Note 110: Environmental Changes: Temperature, Pressure, Humidity. Retrieved from www.alphasense.com, pages 1–6.

*References*

*Schmitz, S., et al. (2021). "Do new bike lanes impact air pollution exposure for cyclists?—a case study from Berlin." Environmental Research Letters 16(8). https://doi.org/10.1088/1748-9326/ac1379.*

---

## Author Comment (AC2)

**Reviewer #2**

**General comments.** The manuscript describes an open source, systematic methodology to calibrate low-cost sensors (LCSs). The Authors propose a 7-step statistical method based on: 1) preliminary analysis of raw data; 2) data cleaning; 3) flag data; 4) selection of the model by using both multiple linear regression and random forest and several statistical parameters; 5) model validation; 6) export of the experimental data as concentrations; 7) error predictions. Finally, the Authors tested the proposed model with an example during a field campaign in urban environment.

The manuscript shows a very interesting and systematic methodology to calibrate LCSs, suggesting to employ a univocal and standardized method to let comparable the LCSs measurements, considering the more and more frequently use of this technology. Despite this, the manuscript requires revisions before to be accepted for final publication. Following suggestions and specific comments.

*We thank the reviewer for their positive overall evaluation of this manuscript. Their comments have improved the quality of this manuscript. What follows are point-by-point responses to their comments regarding recommended changes to the manuscript.*

**Specific comments.**

For this calibration procedure, reference instruments are needed. Trends due to specific events (i.e., burning etc…) could be not properly described by the sensors, if not calibrated in the same conditions?

*As the reviewer points out, this is a general limitation of low-cost sensors, as they do not have the same level of accuracy as reference instruments and are susceptible to various cross-sensitivities. It is likely that the low-cost sensors would respond to specific events (i.e., burning), however if they were not calibrated with reference instruments under conditions in which such events occur, then it would be very challenging to ensure that the applied calibration accurately represented pollutant concentrations under substantially different ambient conditions. Where available, external data from monitoring stations could be used to identify such events and calibrations could be adjusted post-hoc, if need be.*

*Text has been added to section 3.3, which reads, "Further reference data on other species would have been beneficial to this calibration, as the MOS do exhibit cross-sensitivities to other species, but resources were insufficient, and these data were not collected."*

Moreover, did the Authors try to do a calibration procedure by using chemical standard to produce a calibration curve at different concentrations and conditions in laboratory experiment? If yes, could the Authors discuss difference between the two approaches?

*We thank the reviewer for this comment. A laboratory calibration was not conducted in this study as we did not have the technical resources to do so. Furthermore, calibration curves for various species are provided in the technical specifications of the individual sensors and served as a basis for understanding their sensitivity. There are many other studies that have produced calibration curves or tested LCS in both laboratory and field conditions, but as this methodology focuses on the field calibration of LCS, we did not include any discussion of laboratory calibrations. Furthermore, as several studies have shown, laboratory calibrations are often not transferrable to calibration of ambient measurements, see e.g., Castell et al., (2017), Rai et al., (2017).*

In case of LCSs time drift, did the proposed methodologies take into account of it (or allow to)? Is the proposed frequency (2 weeks every 2-3 months) enough to take into account of seasonal variations and, eventually, time drift of the sensors?

*We thank the reviewer for this comment. Sensor drift is indeed a significant challenge for long-term operation of LCS. In our experience, the proposed frequency of co-location is sufficient to account for sensor drift, which generally occurs in significant amounts 4 months after initial calibration (Peterson et al., 2017). As such, the frequency of co-location in this study would account for any significant drift. Between the summer and winter campaigns, drift might have occurred on a significant scale, but co-location data were not used from one campaign to train models in the other, therefore this should not be a problem. This is, however, an important topic that will be explored in future studies with experiments conducted over longer timescales.*

*Text has been added to Section 3.1, which reads, "These MOS sensors typically experience significant amounts of drift four months after initial calibration, which is why in this study co-locations were conducted at high frequency, before and after each experiment."*

Lines 430 and following. LCSs could have a T and RH dependency. Is it appropriate to apply the suggested corrections by the manufacturer for T and RH to the raw data before to apply the models in the proposed methodology? Could the Authors discuss this aspect and if the models relationships with RH and T are in line with what, eventually, suggested by the manufacturer?

*We thank the reviewer for this comment. For these sensors (SGX Sensortech MICS-4514), no direct corrections for T or RH are recommended by the manufacturer. Instead, they provide a recommended range of operation for these meteorological parameters and discuss the conditions under which sensor sensitivity tests were conducted. Other studies (i.e., Peterson et al. (2017)) have identified that MOS sensor performance can be affected by T and RH and therefore we include these parameters in the model selection process. In this study, as can be seen in Tables 2-5, the most accurate models for predicting $NO_2$ and $O_3$ included both T and RH for the MLR models, and either T or RH for the RF models.*

Other species could interfere with measurements: if the concentration of those compounds changes (season, night-day, etc…), this could affect the sensors response. How the Author suggest to deal with this eventuality?

*We thank the reviewer for this comment. Such cross-sensitivities do exist and interfere with the performance of low-cost sensors. The methodology presented here accoutns for these types of changes by recommending relatively frequent co-locations to capture changes in conditions owning to seasonality, etc. and subsequent effects on ambient conditions, including pollutant changes. We recognize however, that we cannot attribute changes in the model to changes in specific compounds that are interfering. To properly account for all of these, the low-cost sensors would need to be rigorously calibrated alongside reference instruments that measure a much wider range of the species to which the sensors are potentially cross-sensitive. This was not possible for this study and would not be for most studies, and therefore the attribution of impact of these species is unfortunately unknown. It is likely that a portion of the uncertainty in these measurements can be attributed to the impact of these cross-sensitivities on sensor performance.*

*To account for changes in chemistry due to daytime and nighttime conditions, we included a binary "Time of Day" variable (see section 3.4.4) in the model selection process, though in most cases it was deemed to be unimportant to prediction of $NO_2$ and $O_3$. A sentence has been amended in*

*section 2.1, which now reads "Regular co-location allows for the establishment of datasets that cover not only changes in meteorology, but also sensor functioning and interactions of potentially cross-sensitive species."*

The Authors showed their experiment results only for NO2 and O3 MOS sensors. Is this methodology applicable to other compounds (i.e., VOCs) or technologies (i.e., PID, electrochemical) with the same characteristics proposed in the manuscript? This information should be included in the manuscript.

*We thank the reviewer for their comment. Reviewer #1 also pointed this out and this has been answered as follows:*

*A brief physical explanation has been added in the text. While different in their design, both MOS and EC sensors produce a measure of voltage which varies in response to changing concentrations of gas-phase species, and hence can be calibrated using the same methodology. In more recent published work, we have successfully applied this methodology for the calibration of EC sensors, see Schmitz et al., (2021).*

*The added text reads "Furthermore, while it was applied here to sensor systems containing metal oxide LCS, this methodology is also equally as applicable to electrochemical LCS or photoionization detectors (PID), as these produce a similar measure of voltage that varies in response to changing concentrations of gas-phase species and have similar cross-sensitivities to temperature and relative humidity. It is not directly applicable for optical particle counters (OPC) for the measurement of particulate matter, as the transformation of the raw data into concentrations during calibration functions differently, though some of the principles discussed here are still relevant. For an application of this methodology to EC sensors, please see Schmitz et al., (2021)."*

The Authors refer to Ammonia and Reducing gas sensor in the manuscript (see Table 1), but results regarding these sensors are not present. Is this due to lacking of reference instrument?

*Yes, this is correct. No reference instrumentation for $NH_3$ was available for this study and the focus was on $NO_2$ and $O_3$.*

Could the Author describe in more details how the Step 4 is performed? How the experiment and co-location data have been used in this step? The Authors describe that in Step 5 and 6 the co-locations data were used, but information about data used in Step 4 seem missed.

*We thank the reviewer for this comment. The same co-location data that is used in model validation in step 5 and in training the final models in step 6 is used in the model selection process of step 4. In this step, the co-location data are broken into multiple continuous training and test data sets upon which various statistical models using combinations of the various LCS and meteorological variables are tested. Statistical parameters of $R^2$, RMSE, MAE, and AIC are used to select models of best fit. Once the best-fitting models are determined, these are then passed on to step 5, the model validation step.*

*Text has been added to section 2.2.4 to clarify that the data used in the model selection process are those from the co-locations. This now reads, "The co-location data were used in this step to train various models and determine the best fitting MLR and RF models."*

Did the Authors intercompare between them similar sensors, i.e. two Zephyrs, before and after the calibration to check the response of same sensors in same conditions?

*We thank the reviewer for this comment. This same issue was raised by reviewer #1. Our response is as follows:*

*For this study, we use the two low-cost sensors primarily as examples of how to use the seven-step methodology and did not consider their intercomparison as we felt it might distract from the main focus of the work. However, we have added graphs depicting the agreement between standardized raw LCS data of s71 and s72 during the 2 co-locations of the winter measurement campaign into the supplemental information (Figures S4 and S5). The Oxa and O3a MOS sensors of each sensor system are linearly related, but due to differences in sensor sensitivity, have different baselines. In the summer campaign, the relationship between the O3a sensors of s71 and s72 during co-location 2 is non-linear but returns to linear agreement in co-location 3 and in the winter campaign.*

*A reference to the added graphs in SI and a brief discussion of this point in the text was added [section 3.3] and reads, "To compare sensor performance between s71 and s72, an intercomparison of available co-location raw data was conducted for the oxidizing MOS (Oxa) and ozone MOS (O3a). During all co-locations, the sensors had a linear relationship and an $R^2 > 0.95$ (Figures S4 and S5). In only one instance was this not the case (co-location 2, O3a), where the $R^2$ was 0.59 and a deviation from linearity was detected. This relationship was linear and normal in all other co-locations.".*

About the data cleaning, how the Authors correct data for possible bias effect? Line 190: the duration of the moving window chosen to remove the outliers avoid to exclude from the dataset some specific and real events with short duration?

*We thank the reviewer for this comment. A similar comment was made by reviewer #1. Our response is as follows:*

*Indeed, this is a significant challenge in data cleaning and unfortunately requires, in some cases, subjective assessment for an accurate determination to be reached. While it is possible that this data cleaning method removes some non-outlier measurements during peak emissions, it is equally possible that such events are indeed outliers due to technical sensor error. For this reason, we recommend that identified outliers be graphically compared with neighboring points to determine if their removal is justified.*

*With the optimized moving window and threshold identified in this study, for sensor system s71, a total of 58 outliers were detected from >500,000 data points. In this case individual assessment of each point's 'outlierness' was practical, but there may be cases where this is impractical. In such cases, we recommend a random subset of outliers be graphically assessed to determine the extent to which the data cleaning function is removing actual outliers. Furthermore, while peak emissions may be misidentified as outliers, these short peaks often occur not as individual points, but as small groups that indicate a peak event. As such, the uppermost point might be removed by this data cleaning method, but those surrounding points would not, ensuring that the peak emissions event remains mostly accounted for. This is imperfect, but it is unlikely that a data cleaning method exists which can perfectly separate outliers from peak events. It is with such peak events that other tested methods such as the AutoRegressive Integrated Moving Average performed particularly poorly, identifying most peak events as outliers. If such events are expected due to the deployment environment, particular care in evaluating the outlier removals should address this.*

*Clarification text has been added to section 2.2.2 regarding this and now reads, "The points flagged as outliers with this method were then graphically assessed against neighboring datapoints to prevent inadvertent removal of peak emission events. In other cases where assessing all outliers is impractical, it is recommended to do so with a random subset of outliers. Furthermore, if substantial short-term events are expected due to the deployment environment, such as during mobile measurements, a more thorough check of the outlier removal should be done."*

Lines 218-220. To identify which model better describe the measurements in term of over or under estimation, could the Authors consider to include also a statistical parameter such as the Fractional Bias?

*We thank the reviewer for this comment. We are unfamiliar with the use of that statistic and while we could find its definition, we could find no information on how to assess the uncertainty on the central estimate due to sample size. Unless the reviewer can point us to a reference that discusses assessment of uncertainty on fractional bias for a given sample size, we have no way of knowing if a particular value derived from our sample actually represents significant bias. At this stage, we ask the reviewer for greater clarification on how they envision the use of fractional bias in this study.*

Lines 365-369. Co-location 3 was at the end of the summer campaign (i.e., October). Anyway data for experiment 2 are not available. It seems from Figure 4 there is a seasonal impact. Did the Author use this co-location for their calibration for Experiment 1? Did the different season affect the calibration procedure? Are the 2 weeks every 2-3 months enough to take into account of it?

*We thank the reviewer for this comment. Indeed, there is a seasonal impact on the data, as the co-location in October experienced cooler temperatures than co-location 2, which took place at the end of the summer. The reason co-location 3 was included in the analysis was to see if the data were still useful for training a model that could predict on experiment 1, which took place in August. As can be seen in tables 8 and 9, the most accurate models for prediction on experiment 1 were those that were trained with co-location 2 exclusively. However, models trained using both co-locations were only slightly more inaccurate. We would have expected co-location 3 to be more relevant for experiment 2, due to the seasonal effects that the reviewer points out. Unfortunately, these hypotheses could not be tested due to loss of data for experiment 2 and co-location 1.*

*Text has been added to section 3.5, in connection with a separate but similar comment. The text reads, "This alludes largely to the fact that seasonal changes caused co-location 3 to experience different meteorological and pollution conditions than were present during the experiment. While results show that this co-location was not useful for accurate prediction, it is likely that it would have been more relevant for prediction on experiment 2, during which the environmental conditions were more comparable. Similarly, co-location 1 would likely have been more valuable for prediction with experiment 1 than with experiment 2. However, due to the loss of data from s71, this could not be assessed more closely in this study."*

Lines 390-396. Is the GSM the only way to transfer data to a database? The warm up time was provided by the manufacturer?

*There are other methods of data transfer, such as through Wi-Fi or a direct data download via cable, but these were not incorporated into the prototype Zephyrs and therefore GSM was the best method for sending data to the external database. The warm-up time and the impact on sensor*

*signal was detected during post-processing and was not provided by the manufacturer but confirmed in subsequent discussions with EarthSense. It was known that the sensors needed time to warm up before stabilizing, but it was not clear that the sensor signal would fluctuate as significantly as it did when the sensors briefly turned off. This further highlights the importance of Step 2 in which the raw data are rigorously cleaned.*

Lines 421-423. Since the 3ʳᵈ co-location is in October, could this be indication that closer and more frequent co-location are needed? See also the following Section 3.5 (line 539) and Figures 14-15.

*We thank the reviewer for this comment. If data from co-location 1 and experiment 2 were available, this would have been assessed more closely in this study. Tables 8 and 9 reveal that inclusion of co-location 3 training data in final models for prediction did not improve predictive accuracy on experiment 1. This reflects in large part the impact of seasonality as the reviewer points out. However, using training data from co-location 2 alone was suitable for prediction on experiment 2 and therefore we do not feel that more frequent co-location was needed in this case. While co-location 3 was not necessary for accurate prediction of experiment 1, it would likely have been necessary for accurate prediction of experiment 2. Therefore, we feel that the spacing of co-locations was appropriate for this study, but the technical failures that led to loss of data for co-location 1 and experiment 2 prevented proper evaluation.*

*Text has been added to section 3.5, in connection with a separate but similar comment. The text reads, "This alludes largely to the impact of seasonal changes on co-location 3, which experienced different meteorological and pollution conditions than were present during the experiment. While results show that this co-location was not useful for accurate prediction, it is likely that it would have been more relevant for prediction on experiment 2, during which the environmental conditions were more comparable. Similarly, co-location 1 would likely have been more valuable for prediction with experiment 1 than with experiment 2. However, due to the loss of data from s71, this could not be assessed more closely in this study."*

Figures 14-15. Could the Authors add the 1:1 lines and indicate the R2 in the plots? How the Authors can explain the constant thresholds in the plots of panels 15e and 15f? Looking at Figure 11, the models using internal T and RH seem to give lower O3 and NO2 compare to the ones that use the ambient T and RH. In figure 11 this is less evident: could the Authors explain it and the reasons/meaning of the slopes (typically lower than the unit) and intercepts?

*We thank the reviewer for this comment. 1:1 lines as well as $R^2$ and RMSE have been added to the plots, which was also requested by reviewer #1.*

*The constant thresholds seen in panels 15e and 15f reflect the nature of the RF models, which cannot predict outside the minimum and maximum values of the co-location data they are trained with. This is a fundamental flaw of RF models and is referred to in the final paragraph of section 3.5. More text has been added here to explain this more clearly.*

*The text reads, "This is a fundamental flaw of RF models as they cannot predict outside the bounds of the co-location data they are trained with.".*

*Regarding the interpretation of slopes and intercepts in figures 14 and 15, a slope greater than 1 indicates overprediction, whereas a slope less than 1 indicates underprediction. The intercepts typically indicate the same (just centered around 0), but in this case, as the models tend to overpredict low concentrations and underpredict high concentrations for $NO_2$, the intercepts are*

*in some cases offset to be positive. For $O_3$ this is less often the case, as the slopes and intercepts more often both agree regarding under or overprediction.*

*To be certain that the models trained with ambient and internal T/RH were giving significantly different concentrations, their predictions were compared to each other using student's t-tests and mann-whitney-wilcoxon U-tests. In all cases the results were statistically significant, confirming the reviewer's suspicions. The likeliest explanation stems from the differences in raw output of the internal and ambient T/RH monitors. These have somewhat non-linear relationships, which in the case of RH becomes especially non-linear under high RH conditions. In fact, the internal T is consistently higher and RH consistently lower than their ambient counterparts. These differences likely lead to the significantly different predictions between ambient and internal MLR and RF models, but the magnitude is not large, as T/RH are less relevant to prediction in all cases than the MOS Oxa/O3a. As follows with previous comments regarding the differences between models trained with internal and ambient T/RH, while the results are interesting and warrant further investigation, we feel that this analysis lies outside the scope of this manuscript and should be the subject of future research.*

*Text that was added to the discussion in line with a previous comment from Reviewer #1 has been modified further, and now reads, "However, given that models using ambient data were more accurate during the validation step and significant differences between predictions of models trained with internal vs ambient T and RH were identified, these results require closer inspection, which should be the subject of future research."*

Supplementary. Why for the winter campaign, the Authors use co-location 1 and 2 instead of 4 and 5, which are closer to Experiment 3? Comparing Table 9 and Table S4, the models identified for O3 are different (and similarly for NO2): how the Authors could explain this?

*We thank the reviewer for this comment and noticing this error. At an earlier stage in the analysis, co-locations 4 and 5 were referred to as 'co-locations 1 and 2' for the winter campaign. The enumeration of the co-locations was not changed in the supplemental information upon submission, but this error has now been corrected.*

**Technical comments.**

Line 88. See "host".

*"…a host of…" has been changed to "many".*

Line 200. Do the Authors refer to Section 3.5?

*Yes, this has been changed.*

Line 291. Decent and good agreement should have to be quantitative and not qualitative information.

*These qualitative statements have been changed and replaced with the actual $R^2$ of the intercomparison between instruments.*

Lines 306-307 and 317-318. Information about the date of the campaigns are confused and should be coherent. The information could be furnished only once clearly and I would suggest to add the dates in Figure 2, as well.

*The dates for the experiments have been moved to the next paragraph and have been placed alongside the dates for the co-locations. Figure 2 has been updated to include the dates.*

Line 328-329. This sentence should be clarified. Zephyr s71 and s72 were located as in Figure 3 or with reference in an office on the 6ᵗʰ floor? In the former case, this information is redundant and could be included in previous paragraphs, when describing the setting (line 311 and following). In the second: how air masses have been sampled?

*This sentence has been moved to the previous paragraph when describing the co-location set-up on the 6ᵗʰ floor of the Mathematics building. Text has been modified in the original paragraph of this sentence for coherence.*

Line 359. When the Authors refer to "combined" co-location, this means an average of co-location 1 and 2?

*Combined co-location refers to the use of both co-locations together, not the average. The text has been updated to be clearer.*

Lines 419-423. This section is not well described. Could the Authors explain in more details the criteria to be used to flag the data?

*We thank the reviewer for this comment. The details regarding the flagging of data are provided previously in section 2.2.3 and therefore were not repeated here. If they feel that the details in section 2.2.3 are insufficient, then we will gladly be more specific.*

Lines 430. Could the Authors specify in this or previous paragraph the units of the input data?

*Done. Units have been added to graphs as well where they were missing.*

Lines 436-438. The Author report that relationship between Oxa and O3 was determined be inverse; but, since the predictive accuracy for no transformation is similar, they selected the latter. Anyway, in Table 3 there is not inverse relationship and a log dependency between O3 and Oxa was selected (also in Table 5 there is not inverse transformation). Could the Author explain this discrepancy or illustrate better this paragraph?

*We thank the reviewer for identifying this discrepancy. The text was not updated alongside new findings during the model selection process, leading to this discrepancy between the text and Table 3. The findings in Table 3 are correct and the text in section 3.4.4 has now been updated accordingly.*

Lines 490-498. A comparison with the reference O3 and NO2 data should be included here (and in Figure 11).

*We thank the reviewer for this comment. An earlier version of this manuscript contained comparisons to $NO_2$ and $O_3$ concentrations in this section but was removed in the submitted version. The justification for this decision was to ensure that in section 3.4, the results were*

*described without reference concentrations as under normal experimental deployment, a comparison of calibrated LCS data to reference instruments would not be possible. Therefore, to keep the example of the use of the methodology more relevant for realistic scenarios, we removed the comparisons to reference concentrations in step 6. The comparison between calibrated data and reference concentrations was then placed exclusively in section 3.5, which describes the atypical extra validation step taken in this study. In this regard, we prefer to not include reference concentrations in step 6, as it is not realistic to expect that the experimental deployment of LCS will occur alongside reference instrumentation in the majority of future studies and applications. If the reviewer feels that the comparisons to reference data made in Tables 8 and 9, as well as in Figures 14 and 15, are insufficient, then we will reconsider including reference data during Step 6.*

*Line 595. See "this is should be".*

*This grammatical error has been fixed.*

**References**

*Castell, N., et al. (2017). "Can commercial low-cost sensor platforms contribute to air quality monitoring and exposure estimates?" Environ Int 99: 293-302. https://doi.org/10.1016/j.envint.2016.12.007.*

*Peterson, P. J. D., et al. (2017). "Practical Use of Metal Oxide Semiconductor Gas Sensors for Measuring Nitrogen Dioxide and Ozone in Urban Environments." Sensors (Basel) 17(7). https://doi.org/10.3390/s17071653.*

*Rai, A. C., et al. (2017). "End-user perspective of low-cost sensors for outdoor air pollution monitoring." Sci Total Environ 607-608: 691-705. https://doi.org/10.1016/j.scitotenv.2017.06.266.*

*Schmitz, S., et al. (2021). "Do new bike lanes impact air pollution exposure for cyclists?—a case study from Berlin." Environmental Research Letters 16(8). https://doi.org/10.1088/1748-9326/ac1379.*